# MATRIX AND TENSOR COMPLETION WITH NOISE VIA LOW-RANK DECONVOLUTION

## ABSTRACT

Low-rank Deconvolution (LRD) has been recently introduced as a new representation model for multi-dimensional data. In this work we consider its use for tackling the problem of matrix and tensor completion. This model is designed to encode information in a very efficient manner using a limited number of parameters and to be flexible by providing a simple framework that allows for easy inclusion of priors such different types of regularizers while, at the same time, letting for easy algorithms to solve the (generally non-convex) learning process. We suspect that these properties will facilitate the resolution of tensor completion problems, *i.e.* the reconstruction of a tensor from incomplete and randomly corrupted entries. Then, our contribution is twofold: first we show that this model acts as a relaxation of the classical low-rank approach allowing for a greater number of solutions than the imposed by the low-rank constraint while using a similar number of parameters. And second, we present an algorithm based on a block multi-convex optimization method with nuclear norm minimization and squared total variation regularization that solves the tensor completion problem. Theoretical and empirical results are presented that support our claims.

## 1 INTRODUCTION

Estimating latent models from incomplete and corrupted observations is a fundamental task that covers a diverse range of applications. Matrix completion represents a standard problem of this kind and comes up in many areas of science and engineering, including collaborative filtering (Goldberg et al., 1992; Rennie & Srebro, 2005; Srebro & Salakhutdinov, 2010), control (Liu & Vandenberghe, 2010), remote sensing (Schmidt, 1986), chemometrics (Bro, 1999), integrated radar and communications (Sodagari et al., 2012), system identification (Liu et al., 2013) and computer vision (Chen & Suter, 2004; Tomasi & Kanade, 1992; Ji et al., 2010; Bertalmio et al., 2000), to name a few.

Regarding this problem a wide variety of studies about its statistical performance have been published during the last couple of decades (Candes & Plan, 2010; Candes & Recht, 2012; Chen et al., 2020; Chi & Li, 2019; Li et al., 2019). Generally considering low-rank models as this structure is often present in the nature of these problems, namely low-rank matrix completion (LRMC). This problem can be formulated as:

$$\min_{\mathbf{Z}} \frac{1}{2} \|\mathcal{P}_\Omega(\mathbf{Y} - \mathbf{Z})\|_F^2 \tag{1}$$
$$\text{subject to} \quad \mathbf{Z} = \mathbf{VS},$$

where $\mathbf{Y} \in \mathbb{R}^{n \times m}$ is the measurement matrix, $\mathbf{Z} \in \mathbb{R}^{n \times m}$ is the matrix to be recovered of low-rank $r$, $\mathbf{V} \in \mathbb{R}^{n \times r}$ and $\mathbf{S} \in \mathbb{R}^{r \times m}$ are matrices of low-rank $r$ and $\mathcal{P}_\Omega$ denotes the projection onto the set of known values $\Omega$. This matrix completion problem easily extends to higher dimensional spaces as tensor completion. Concerning that situation, the scientific community has given much of the attention to its computational challenge and its statistical performance (Acar et al., 2011; Barak & Moitra, 2016; Xia et al., 2021), but in general without variations on the latent model, which is assumed to be a LR tensor. And the problem is equivalently low-rank tensor completion (LRTC) with ubiquitous application in different areas such computer vision (Li & Li, 2010; Liu et al., 2012), signal processing (Lim & Comon, 2010; Nion & Sidiropoulos, 2010) and machine learning (Collins & Cohen, 2012; Chaganty & Liang, 2013).

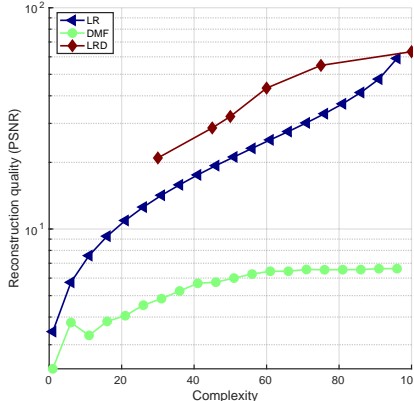

Figure 1: Performance evaluation of matrix reconstruction on city and fruit testing dataset.

More recently deep matrix factorization (DMF) has appeared in the machine learning community as a new latent model for the problem of matrix completion (Fan & Cheng, 2018; Arora et al., 2019). With a general formulation given by,

$$\mathbf{Z} = g_1(\mathbf{X}_1 g_2(\mathbf{X}_2 \cdots g_{L-1}(\mathbf{X}_{L-1}\mathbf{X}_L)\cdots)), \tag{2}$$

where $\{g_i\}_{l=1}^{L-1}$ are activation functions and $\{\mathbf{X}_i\}_{l=1}^{L-1}$ their correspondent activation maps. This model has direct extension to the tensor completion case (Fan, 2022). And similarly to this approach Low-rank Deconvolution (LRD) came out as a new model for tensor data representation (Reixach, 2023; Humbert et al., 2021). It relays on a set of filters convoluted with a set of low-rank and sparse activation maps that can be formulated as:

$$\mathcal{Z} = \sum_m \mathcal{D}_m * \mathcal{K}_m = \sum_m \mathcal{D}_m * [\![\mathbf{X}_m^{(1)}, \ldots, \mathbf{X}_m^{(N)}]\!], \tag{3}$$

where $\mathcal{Z} \in \mathbb{R}^{I_1 \times I_2 \times \cdots \times I_N}$ is the signal to be recovered, $\mathcal{D}_m \in \mathbb{R}^{L_1 \times L_2 \times \cdots \times L_N}$ acts as a dictionary, and $\mathcal{K}_m \in \mathbb{R}^{I_1 \times I_2 \times \cdots \times I_N}$, the activation map, is a low-rank factored tensor (*i.e.* a Kruskal tensor) and can be expressed in terms of its constituent matrices $\{\mathbf{X}_m^{(n)}\}$ as explained in Section 1.1. Then, a generic method for solving the tensor completion problem can be formulated as,

$$\underset{\{\mathbf{X}_m^{(n)}, \mathbf{D}_m\}}{\arg\min} \frac{1}{2} \left\| \mathcal{P}_\Omega(\mathcal{Y} - \sum_{m=1}^M \mathcal{D}_m * [\![\mathbf{X}_m^{(1)}, \ldots, \mathbf{X}_m^{(N)}]\!]) \right\|_2^2 + \sum_{m=1}^M \sum_{n=1}^N \lambda \left\| \mathbf{X}_m^{(n)} \right\|_F^2. \tag{4}$$

In this work we will show how the classical LR model is a special case of LRD with $m = 1$ and $D_m$ the identity filter. Therefore, LRD acts as a relaxation of the LR model. To begin with this concept, we perform an experiment: we reconstruct the city and fruit testing dataset from Zeiler et al. (2010), without missing data and without corruption, using LR (1), DMF (Fan, 2022), and LRD. Results are presented in figure 1. We can observe how, for the same given complexity, LRD is capable of a reconstruction of a greater quality than the other two methods, a hypothesis that will be later proven. In this work we will also show how the LRD model facilitates the development of tensor completion algorithms which can contemplate total variation and nuclear norm regularization allowing to work with corrupted data.

The structure of this paper is as follows: section 1.1 introduces the notation used, section 2 provides theoretical guarantees for the matrix and tensor completion problems using LRD and LR, section 3 presents the tensor completion method based on LRD, section 4 presents the extensive numerical results that verify our claims and section 5 concludes the work.

## 1.1 NOTATION

Let $\mathcal{K} \in \mathbb{R}^{I_1 \times I_2 \times \ldots \times I_N}$ be a $N$-order tensor. The PARAFAC (Harshman et al., 1970) decomposition (a.k.a. CANDECOMP Carroll & Chang (1970)) is defined as:

$$\mathcal{K} \approx \sum_{r=1}^{R} \mu_r \mathbf{v}_r^{(1)} \circ \mathbf{v}_r^{(2)} \circ \ldots \circ \mathbf{v}_r^{(N)}, \tag{5}$$

where $\mathbf{v}_r^{(n)} \in \mathbb{R}^{I_n}$ with $n = \{1, \ldots, N\}$ and $\mu_r \in \mathbb{R}$ with $r = \{1, \ldots, R\}$, represent an one-order tensor and a weight coefficient, respectively. $\circ$ denotes an outer product of vectors. Basically, Eq. equation 5 is a rank-$R$ decomposition of $\mathcal{K}$ by means of a sum of $R$ rank-1 tensors. If we group all these vectors per mode $(n)$, as $\mathbf{X}^{(n)} = \left[ \mathbf{v}_1^{(n)}, \mathbf{v}_2^{(n)}, \ldots, \mathbf{v}_R^{(n)} \right]$, we can define the Kruskal operator (Kolda, 2006) as follows:

$$[\![ \mathbf{X}^{(1)}, \mathbf{X}^{(2)}, \ldots, \mathbf{X}^{(N)} ]\!] = \sum_{r=1}^{R} \mathbf{v}_r^{(1)} \circ \mathbf{v}_r^{(2)} \circ \ldots \circ \mathbf{v}_r^{(N)}, \tag{6}$$

being the same expression as Eq. equation 5 with $\mu_r = 1$ for $\forall r$, i.e., depicting a rank-$R$ decomposable tensor.

For later computations, we also define a matricization transformation to express tensors in a matrix form. Particularly, we will use an special case of matricization known as $n$-mode matricization (Bader & Kolda, 2006; Kolda, 2006). To this end, let $\mathcal{C} = \{c_1, \ldots, c_G\} = \{1, \ldots, n-1, n+1, \ldots, N\}$ be the collection of ordered modes different than $n$, and $\Lambda = \prod_t I_t / I_n$ be the product of its correspondent dimensions; we can express then tensor $\mathcal{K}$ in a matricized array as $^{(n)}\mathbf{K} \in \mathbb{R}^{I_n \times \Lambda}$. Note that we represent the $n$-mode matricization by means of a left super-index. The $n$-mode matricization is a mapping from the indices of $\mathcal{K}$ to those of $^{(n)}\mathbf{K}$, defined as:

$$\left( ^{(n)}\mathbf{K} \right)_{i_n, j} = \mathcal{K}_{i_1, i_2, \ldots, i_N}, \tag{7}$$

with:

$$j = 1 + \sum_{g=1}^{G} \left[ (i_{c_g} - 1) \prod_{g'=1}^{G-1} I_{c_{g'}} \right]. \tag{8}$$

With these ingredients, and defining $\mathcal{J}(\mathbf{X}^{(1)}, \ldots, \mathbf{X}^{(N)}) = [\![ \mathbf{X}^{(1)}, \ldots, \mathbf{X}^{(N)} ]\!]$, we can obtain the $n$-mode matricization of the Kruskal operator as:

$$^{(n)}\mathbf{J} = \mathbf{X}^{(n)} (\mathbf{Q}^{(n)})^\top, \tag{9}$$

with:

$$\mathbf{Q}^{(n)} = \mathbf{X}^{(N)} \odot \ldots \odot \mathbf{X}^{(n+1)} \odot \mathbf{X}^{(n-1)} \odot \ldots \odot \mathbf{X}^{(1)}, \tag{10}$$

where $\odot$ denotes the Khatri-Rao product (Kolda, 2006).

Finally, we can express the vectorized version of Eq. equation 9 as:

$$\text{vec}\left( ^{(n)}\mathbf{J} \right) = \left[ \mathbf{Q}^{(n)} \otimes \mathbf{I}_{I_n} \right] \text{vec}(\mathbf{X}^{(n)}), \tag{11}$$

where $\otimes$ indicates the Kronecker product, and $\text{vec}(\cdot)$ is a vectorization operator. It is worth noting that doing so, the vectorized form of the Kruskal operator is represented by a linear expression.

## 2 THEORETICAL BOUNDS FOR MATRIX AND TENSOR COMPLETION PROBLEMS.

LRD can outperform LR in representation tasks as shown in Section 1. But does it always happen? In this section we provide a theoretical guarantee for LRD that allows to compare it to classical LR methods.

## 2.1 LRD VS. LR IN MATRIX COMPLETION.

**Assumption 2.1.** Suppose $\mathbf{Y} = \mathbf{Z} + \mathbf{E}$, where $\mathbf{Y} \in \mathbb{R}^{I_2 \times I_1}$. $\mathbf{Z} = \sum_{m=1}^{M} \mathbf{F}_2 \, {}^{(2)}\hat{\mathbf{D}}_m \oplus \hat{\mathbf{X}}_m^{(2)}(\hat{\mathbf{X}}_m^{(1)})^T \mathbf{F}_1$ is generated by the matricized form of the LRD model and formulated in the DFT domain (for more details see Section 3), where $\hat{\mathbf{X}}_m^{(n)} \in \mathbb{C}^{I_n \times r}$, $\mathbf{F}_1 \in \mathbb{C}^{I_1 \times I_1}$, $\mathbf{F}_2 \in \mathbb{C}^{I_2 \times I_2}$, ${}^{(2)}\hat{\mathbf{D}}_m \in \mathbb{C}^{I_2 \times I_1}$ and $\oplus$ denotes hadamard product. The entries of the noise matrix $\mathbf{E}$ are drawn from $\mathcal{N}(0, \sigma^2)$. Suppose we observed a few entries of $\mathbf{Y}$ randomly (sampling without replacement) and want to recover $\mathbf{Z}$ from the incomplete $\mathbf{Y}$. Consider the following method:

$$\min_{\{\hat{\mathbf{X}}_m^{(n)}, {}^{(2)}\hat{\mathbf{D}}_m\}} \frac{1}{2}\|\mathcal{P}_\Omega(\mathbf{Y} - \sum_{m=1}^{M} \mathbf{F}_2 \, {}^{(2)}\hat{\mathbf{D}}_m \oplus \hat{\mathbf{X}}_m^{(2)}(\hat{\mathbf{X}}_m^{(1)})^T \mathbf{F}_1)\|_F^2 + \frac{\mu_X}{2}\sum_{m=1}^{M}\sum_{n=1}^{2}\|\hat{\mathbf{X}}_m^{(n)}\|_F^2$$

$$+ \frac{\mu_D}{2}\sum_{m=1}^{M}\|{}^{(2)}\hat{\mathbf{D}}_m\|_F^2 \tag{12}$$

Where $\mu_X$ and $\mu_D$ are regularization parameters and it can be shown that matches the matricized version of (4) for an appropriate choice of $\mu_D$. The following theorem provides the excess risk bound for (12).

**Theorem 2.2.** *Suppose* $\{\hat{\mathbf{X}}_m^{(1)}\}_{m=1}^M, \{\hat{\mathbf{X}}_m^{(2)}\}_{m=1}^M, \{{}^{(2)}\hat{\mathbf{D}}_m\}_{m=1}^M$ *are given by (12). Let* $\hat{\mathbf{Z}} = \sum_{m=1}^{M} \mathbf{F}_2 \, {}^{(2)}\hat{\mathbf{D}}_m \oplus \hat{\mathbf{X}}_m^{(2)}(\hat{\mathbf{X}}_m^{(1)})^T \mathbf{F}_1$. *Suppose* $\max(\|\mathbf{Y}\|_\infty, \|\hat{\mathbf{Z}}\|_\infty) \leq \xi$ *and* $I_\pi = \prod_{i=1}^{n} I_i$. *Then with probability at least* $1 - 2I_\pi^{-1}$:

$$\frac{1}{\sqrt{I_\pi}}\|\mathbf{Z} - \hat{\mathbf{Z}}\|_F = \frac{1}{\sqrt{|\Omega|}}\|P_\Omega(\mathbf{Y} - \hat{\mathbf{Z}})\|_F + \frac{1}{\sqrt{I_\pi}}\|\mathbf{E}\|_F + \frac{2\epsilon}{\sqrt{|\Omega|}}$$

$$+ \left(\frac{8\xi^4(\log(I_\pi) + (I_1 r + 2I_\pi)\log(\frac{3^4 M^3 \sum_{m=1}^M (\beta_m^0 \beta_m^1 \beta_m^2)^3}{\epsilon}))}{|\Omega|}\right)^{1/4} \tag{13}$$

*Proof.* See appendix C.2 $\qquad\qquad\square$

Now observe how a classical LR method is a specific case of a LRD with an identity dictionary (*i.e.* ${}^{(2)}\hat{\mathbf{D}} = \mathbf{1}$).

**Assumption 2.3.** Suppose $\mathbf{Y} = \mathbf{Z} + \mathbf{E}$, where $\mathbf{Y} \in \mathbb{R}^{I_2 \times I_1}$. $\mathbf{Z} = \mathbf{X}^{(2)}(\mathbf{X}^{(1)})^T$ is generated by the matricized form of the LR model. Dismissing the DFT matrices by considering directly the spatial versions $\mathbf{X}^{(n)}$, the classical LR method can be formulated as:

$$\min_{\{\mathbf{X}^{(n)}\}} \frac{1}{2}\|\mathcal{P}_\Omega(\mathbf{Y} - \mathbf{X}^{(2)}(\mathbf{X}^{(1)})^T)\|_F^2 + \frac{\mu_X}{2}\sum_{n=1}^{2}\|\mathbf{X}^{(n)}\|_F^2 \tag{14}$$

With this consideration, we obtain the following theorem, that provides the excess risk bound for classical LR.

**Theorem 2.4.** *Suppose* $\mathbf{X}^{(1)}, \mathbf{X}^{(2)}$, *are given by (14). Let* $\hat{\mathbf{Z}} = \mathbf{X}^{(2)}(\mathbf{X}^{(1)})^T$. *Suppose* $\max(\|\mathbf{Y}\|_\infty, \|\hat{\mathbf{Z}}\|_\infty) \leq \xi$. *With* $I_\pi$ *defined above. Then with probability at least* $1 - 2I_\pi^{-1}$:

$$\frac{1}{\sqrt{I_\pi}}\|\mathbf{Z} - \hat{\mathbf{Z}}\|_F = \frac{1}{\sqrt{|\Omega|}}\|P_\Omega(\mathbf{Y} - \hat{\mathbf{Z}})\|_F + \frac{1}{\sqrt{I_\pi}}\|\mathbf{E}\|_F + \frac{2\epsilon'}{\sqrt{|\Omega|}} +$$

$$\left(\frac{8\xi^4(\log(I_\pi) + (I_1 r + I_\pi)\log(\frac{12(\beta^1\beta^2)^2}{\epsilon'}))}{|\Omega|}\right)^{1/4} \tag{15}$$

*Proof.* See appendix C.3 $\qquad\qquad\square$

As classical LR is limited to $^{(2)}\hat{\mathbf{D}} = \mathbf{1}$ we have $\epsilon' >> \epsilon$ when the latent model includes a dictionary different that the identity. If we compare (13) and (15) we observe how the third term is greater in the second case while the fourth is smaller. However, the minorization of the fourth term happens inside a logarithm operator which allows us to conclude that LRD provides a tighter generalization bound than the classical LR when the latent model is not strictly LR.

## 2.2 LRD vs LR in Tensor Completion.

Now we consider the same as in section 2.1 but for three-dimensional tensors.

**Assumption 2.5.** Suppose $\mathcal{Y} = \mathcal{Z} + \mathcal{E}$, where $\mathcal{Y} \in \mathbb{R}^{I_1 \times I_2 \times I_3}$. $\mathbf{Z} = \sum_{m=1}^{M} \mathbf{F}_3 \,^{(3)}\hat{\mathbf{D}}_m \oplus \hat{\mathbf{X}}_m^{(3)}(\hat{\mathbf{X}}_m^{(1)} \odot \hat{\mathbf{X}}_m^{(2)})^T \mathbf{F}_{12}$ is generated by the matricized form of the LRD model and formulated in the DFT domain (for more details see Section 3), where $\hat{\mathbf{X}}_m^{(n)}$ is defined above, $\mathbf{F}_{12} \in \mathbb{C}^{I_1 I_2 \times I_1 I_2}$, $\mathbf{F}_3 \in \mathbb{C}^{I_3 \times I_3}$, $^{(3)}\hat{\mathbf{D}}_m \in \mathbb{C}^{I_3 \times I_1 I_2}$. The entries of the noise tensor $\mathcal{E}$ are drawn from $\mathcal{N}(0, \sigma^2)$. Suppose we observed a few entries of $\mathbf{Y}$ randomly (sampling without replacement) and want to recover $\mathbf{Z}$ from the incomplete $\mathbf{Y}$. Consider the following method:

$$\min_{\{\hat{\mathbf{X}}_m^{(n)}, \,^{(3)}\hat{\mathbf{D}}_m\}} \frac{1}{2}\|\mathcal{P}_\Omega(\mathbf{Y} - \sum_{m=1}^{M} \mathbf{F}_3 \,^{(3)}\hat{\mathbf{D}}_m \oplus \hat{\mathbf{X}}_m^{(3)}(\hat{\mathbf{X}}_m^{(1)} \odot \hat{\mathbf{X}}_m^{(2)})^T \mathbf{F}_{12}\|_F^2 + \frac{\mu_X}{2} \sum_{m=1}^{M} \sum_{n=1}^{3} \|\hat{\mathbf{X}}_m^{(n)}\|_F^2$$

$$+ \frac{\mu_D}{2} \sum_{m=1}^{M} \|^{(3)}\hat{\mathbf{D}}_m\|_F^2 \tag{16}$$

Which can be shown that matches the matricized version of (4) for an appropriate choice of $\mu_D$. The following theorem provides the excess risk bound for (16).

**Theorem 2.6.** *Suppose* $\{\hat{\mathbf{X}}_m^{(1)}\}_{m=1}^{M}, \{\hat{\mathbf{X}}_m^{(2)}\}_{m=1}^{M}, \{\hat{\mathbf{X}}_m^{(3)}\}_{m=1}^{M}, \{^{(3)}\hat{\mathbf{D}}_m\}_{m=1}^{M}$ *are given by (16). Let* $\hat{\mathbf{Z}} = \sum_{m=1}^{M} \mathbf{F}_3 \,^{(3)}\hat{\mathbf{D}}_m \oplus \hat{\mathbf{X}}_m^{(3)}(\hat{\mathbf{X}}_m^{(1)} \odot \hat{\mathbf{X}}_m^{(2)})^T \mathbf{F}_{12}$. *Suppose* $\max(\|\mathbf{Y}\|_\infty, \|\hat{\mathbf{Z}}\|_\infty) \leq \xi$. *With* $I_\pi$ *defined above. Then with probability at least* $1 - 2I_\pi^{-1}$:

$$\frac{1}{\sqrt{I_\pi}}\|\mathbf{Z} - \hat{\mathbf{Z}}\|_F = \frac{1}{\sqrt{|\Omega|}}\|P_\Omega(\mathbf{Y} - \hat{\mathbf{Z}})\|_F + \frac{1}{\sqrt{I_\pi}}\|\mathbf{E}\|_F + \frac{2\epsilon}{\sqrt{|\Omega|}}$$

$$+ \left(\frac{8\xi^4(\log(I_\pi) + ((1 + I_2)I_1 r + 2I_\pi)\log(\frac{3 \cdot 4^4 M^4 \sum_{m=1}^{M}(\beta_m^0 \beta_m^1 \beta_m^2 \beta_m^3 k_r)^4}{\epsilon}))}{|\Omega|}\right)^{1/4} \tag{17}$$

*Proof.* See appendix C.4 □

Again, we observe how a classical LR method is a specific case of a LRD with an identity dictionary (*i.e.* $^{(3)}\hat{\mathbf{D}} = \mathbf{1}$) also for the three-dimensional case. We give the following assumption:

**Assumption 2.7.** Suppose $\mathcal{Y} = \mathcal{Z} + \mathcal{E}$, where $\mathcal{Y} \in \mathbb{R}^{I_1 \times I_2 \times I_3}$. $\mathbf{Z} = \mathbf{X}^{(3)}(\mathbf{X}^{(1)} \odot \mathbf{X}^{(2)})^T$ is generated by the matricized form of the LR model. And, dismissing the DFT matrices by considering directly the spatial versions $\mathbf{X}^{(n)}$, the classical LR method can be formulated as:

$$\min_{\{\mathbf{X}^{(n)}\}} \frac{1}{2}\|\mathcal{P}_\Omega(\mathbf{Y} - \mathbf{X}^{(3)}(\mathbf{X}^{(1)} \odot \mathbf{X}^{(2)})^T)\|_F^2 + \frac{\mu_X}{2} \sum_{n=1}^{2} \|\mathbf{X}^{(n)}\|_F^2 \tag{18}$$

With that in mind we obtain the following theorem, that provides the excess risk bound for classical LR for the three-dimensional case.

**Algorithm 1 LRD.** Solves Eq. equation 20 as an alternating optimization between the code and the dictionary. The code update step is done by means of the LRD method (Alg. 2). The dictionary update step is done by a masked application of the ADMM method from Wohlberg (2015).

1: **Input:** $\mathcal{Y}, \{\mathcal{D}_{0,m}\}_{m=1}^M, R > 0, \Omega$
2: **Output:** $\{\mathcal{D}_{,m}\}_{m=1}^M$
3: **while** not converged **do**
4:     $\{\mathbf{X}_m^{(n)}\} = \arg\min \frac{1}{2} \left\| \mathcal{P}_\Omega(\mathcal{Y} - \sum_{m=1}^M \mathcal{D}_m * [\![\mathbf{X}_m^{(1)}, \ldots, \mathbf{X}_m^{(N)}]\!]) \right\|_2^2 + \Phi(\{\mathbf{X}_m^{(n)}\})$          $\triangleright$ Code Update
5:     $\{\mathcal{D}_m\} = \arg\min \frac{1}{2} \left\| \mathcal{P}_\Omega(\mathcal{Y} - \sum_{m=1}^M \mathcal{D}_m * [\![\mathbf{X}_m^{(1)}, \ldots, \mathbf{X}_m^{(N)}]\!]) \right\|_2^2$          $\triangleright$ Dictionary Update
        s. t.    $\mathcal{D}_m \in C \ \forall m$
6: **end while**

**Theorem 2.8.** *Suppose* $\mathbf{X}^{(1)}, \mathbf{X}^{(2)}, \mathbf{X}^{(3)}$, *are given by* (18). *Let* $\hat{\mathbf{Z}} = \mathbf{X}^{(3)}(\mathbf{X}^{(1)} \odot \mathbf{X}^{(2)})^T$. *Suppose* $\max(\|\mathbf{Y}\|_\infty, \|\hat{\mathbf{Z}}\|_\infty) \leq \xi$. *With* $I_\pi$ *defined above. Then with probability at least* $1 - 2I_\pi^{-1}$:

$$\frac{1}{\sqrt{I_\pi}}\|\mathbf{Z} - \hat{\mathbf{Z}}\|_F = \frac{1}{\sqrt{|\Omega|}}\|P_\Omega(\mathbf{Y} - \hat{\mathbf{Z}})\|_F + \frac{1}{\sqrt{I_\pi}}\|\mathbf{E}\|_F + \frac{2\epsilon'}{\sqrt{|\Omega|}} +$$

$$\left( \frac{8\xi^4(\log(I_\pi) + ((1 + I_2)I_1 r + I_\pi)\log(\frac{3^4(\beta^1\beta^2\beta^3 k_r)^3}{\epsilon'})}{|\Omega|} \right)^{1/4} \tag{19}$$

*Proof.* See appendix C.5                                                                                    $\square$

If we compare (17) and (19), we observe again the same behaviour as in section 2.1, which allows us to conclude that LRD provides a tighter generalization bound than the classical LR when the latent model is not strictly LR.

# 3 LOW-RANK DECONVOLUTION

We now derive the formulation of LRD for our problem. Let $\mathcal{Y} \in \mathbb{R}^{I_1 \times I_2 \times \cdots \times I_N}$ be a multidimensional signal. Our goal is to obtain a multidimensional convolutional representation $\mathcal{Y} \approx \sum_m \mathcal{D}_m * \mathcal{K}_m$, where $\mathcal{D}_m \in \mathbb{R}^{L_1 \times L_2 \times \cdots \times L_N}$ acts as a dictionary, and $\mathcal{K}_m \in \mathbb{R}^{I_1 \times I_2 \times \cdots \times I_N}$, the activation map, is a low-rank factored tensor (*i.e.* a Kruskal tensor). If we write $\mathcal{K}_m = [\![\mathbf{X}_m^{(1)}, \mathbf{X}_m^{(2)}, \ldots, \mathbf{X}_m^{(N)}]\!]$ with $\mathbf{X}_m^{(n)} \in \mathbb{R}^{I_n \times R}$, we can obtain a non-convex problem as:

$$\arg\min_{\mathbf{X}_m^{(n)}, \mathbf{D}_m^{(n)}} \frac{1}{2} \left\| \mathcal{P}_\Omega(\mathcal{Y} - \sum_{m=1}^M \mathcal{D}_m * [\![\mathbf{X}_m^{(1)}, \ldots, \mathbf{X}_m^{(N)}]\!]) \right\|_F^2 + \Phi(\{\mathbf{X}_m^{(n)}\}) \tag{20}$$

where $*$ indicates a $N$-dimensional convolution. Following Reixach (2023) we propose to solve the problem alternating first between the activation map (Kruskal tensor) and the dictionary, and for the activation maps case, again for each Kruskal factor $(n)$ alternately, Algorithm 1 and appendix A provide with more details.

## 3.1 CODE UPDATE STEP

$$\arg\min_{\mathbf{X}_m^{(n)}} \mathcal{L}(\mathbf{X}_m^{(n)})$$

$$\mathcal{L}(\mathbf{X}_m^{(n)}) = \frac{1}{2} \left\| \mathcal{P}_\Omega(\mathcal{Y} - \sum_{m=1}^M \mathcal{D}_m * [\![\mathbf{X}_m^{(1)}, \ldots, \mathbf{X}_m^{(N)}]\!]) \right\|_F^2 + \Phi(\{\mathbf{X}_m^{(n)}\}) \tag{21}$$

Following Reixach (2023) we solve the LRD optimization problems in a DFT domain assuming that boundary effects are neglegible (*i.e.* relying on the use of filters of small spatial support). To this end, we denote by $\hat{\mathbf{A}}$ an arbitrary variable $\mathbf{A}$ in the DFT domain. Let $\hat{\mathbf{D}}_m^{(n)} = \text{diag}(\text{vec}(^{(n)}\hat{\boldsymbol{D}}_m)) \in$

$\mathbb{R}^{\Lambda I_n \times \Lambda I_n}$ be a linear operator for computing convolution, and $\hat{\mathbf{x}}_m^{(n)} = \text{vec}(\hat{\mathbf{X}}_m^{(n)}) \in \mathbb{R}^{RI_n}$ be the vectorized Kruskal factor. We define $\hat{\mathbf{Q}}_m^{(n)} = \hat{\mathbf{X}}_m^{(N)} \odot \cdots \odot \hat{\mathbf{X}}_m^{(n+1)} \odot \hat{\mathbf{X}}_m^{(n-1)} \odot \cdots \odot \hat{\mathbf{X}}_m^{(1)} \in \mathbb{R}^{\Lambda \times R}$, as it was done in Eq. 10, with $\Lambda$ defined in section 1.1. We also define the following supporting expressions:

$$\hat{\mathbf{W}}_m^{(n)} = \hat{\mathbf{D}}_m^{(n)} \big[ \hat{\mathbf{Q}}_m^{(n)} \otimes \mathbf{I}_{I_n} \big], \tag{22}$$

$$\hat{\mathbf{W}}^{(n)} = \big[ \hat{\mathbf{W}}_0^{(n)}, \hat{\mathbf{W}}_1^{(n)}, \dots, \hat{\mathbf{W}}_M^{(n)} \big], \tag{23}$$

$$\hat{\mathbf{x}}^{(n)} = \big[ (\hat{\mathbf{x}}_0^{(n)})^\top, (\hat{\mathbf{x}}_1^{(n)})^\top, \dots, (\hat{\mathbf{x}}_M^{(n)})^\top \big]^\top. \tag{24}$$

All these algebraic modifications allow us to formulate the optimization problem masking out the unknowns which are given in the spatial domain. This requires us to include the DFT transform matrix in our formulation and to consider the spatial version of the signal $\mathbf{y}^{(n)}$ which correspond to the vectorized version of $^{(n)}\mathbf{Y}$:

$$\arg \min_{\hat{\mathbf{x}}^{(n)}} \frac{1}{2} \left\| \mathbf{y}^{(n)} - \mathbf{P}^{(n)} \hat{\mathbf{F}}^{(n)} \hat{\mathbf{W}}^{(n)} \hat{\mathbf{x}}^{(n)} \right\|_2^2 + \Phi(\{\hat{\mathbf{x}}_m^{(n)}\}). \tag{25}$$

Here, $\hat{\mathbf{F}}^{(n)} = \hat{\mathbf{F}}_N \otimes \cdots \otimes \hat{\mathbf{F}}_{n+1} \otimes \hat{\mathbf{F}}_{n-1} \otimes \cdots \otimes \hat{\mathbf{F}}_1 \otimes \hat{\mathbf{F}}_n$ is the matricization of the multilinear DFT inverse transform being $\hat{\mathbf{F}}_i$ the inverse transform for mode-$i$ and $\mathbf{P}^{(n)}$ the mask matrix, which is diagonal for a tensor completion problem. Then its solution will depend on the regularizer chosen $(\Phi(\{\hat{\mathbf{x}}_m^{(n)}\}))$.

## 3.2 Plug-and-Play Total Variation and Nuclear Norm Regularization

The idea of Plug-and-Play regularization (Venkatakrishnan et al., 2013) is to consider a prior model for data reconstruction to be included in the optimization procedure. Following the approach by Rudin et al. (1992), a squared Total Variation and a Nuclear Norm regularizer can be added to the global problem in the following manner:

$$\arg \min_{\{\mathbf{X}_m^{(n)}\}, \boldsymbol{\mathcal{U}}} \frac{1}{2} \|\mathcal{P}_\Omega(\boldsymbol{\mathcal{Y}} - \boldsymbol{\mathcal{U}})\|_2^2 + \frac{\gamma}{2} \|\boldsymbol{\mathcal{U}}\|_{TV}^2 + \lambda \sum_m \left\| \hat{\mathbf{X}}_m^{(n)} \right\|_{NN} + \Psi(\{\mathbf{X}_m^{(n)}\}),$$

$$\text{subject to} \quad \boldsymbol{\mathcal{U}} = \sum_{m=1}^M \boldsymbol{\mathcal{D}}_m * [\![ \mathbf{X}_m^{(1)}, \dots, \mathbf{X}_m^{(N)} ]\!] \tag{26}$$

where $\|\cdot\|_{NN}$ denotes the nuclear norm operator. Now, with a little algebraic manipulation, and making use of the derivative property of the DFT, we can present the following proposition:

**Proposition 3.1.** *The problem presented in eq. (26) is equivalent to problem of eq. (21) with*

$$\Phi(\{\mathbf{X}_m^{(n)}\}) = \frac{\gamma}{2} \left\| (\hat{\Theta}^{(n)})^T \hat{\mathbf{x}}^{(n)} \right\|_2^2 + \lambda \sum_m \left\| \hat{\mathbf{X}}_m^{(n)} \right\|_{NN} + \Psi(\{\mathbf{X}_m^{(n)}\}). \tag{27}$$

*With $\Psi(\{\mathbf{X}_m^{(n)}\}) = \sum_{m=1}^M \sum_{n=1}^N \frac{\alpha}{2} \left\| \mathbf{X}_m^{(n)} \right\|_2^2$ it can be solved by means of gradient descent with a gradient given by*

$$\nabla \mathcal{L}(\hat{\mathbf{x}}_m^{(n)}) = \big[ (\hat{\mathbf{T}}_m^{(n)})^H \hat{\mathbf{T}}_m^{(n)} + \gamma (\hat{\Theta}^{(n)})_m^H \hat{\Theta}_m^{(n)} + \alpha \mathbf{I}_\beta \big] \hat{\mathbf{x}}^{(n)} - (\hat{\mathbf{T}}_m^{(n)})^H \hat{\mathbf{y}}^{(n)} + \lambda \hat{\mathbf{C}}_m^{(n)} (\hat{\mathbf{A}}_m^{(n)})^T, \tag{28}$$

*where $\hat{\mathbf{T}}_m^{(n)} = \mathbf{P}^{(n)} \hat{\mathbf{F}}^{(n)} \hat{\mathbf{W}}_m^{(n)}$ and $\hat{\mathbf{y}}^{(n)}, \hat{\mathbf{x}}_m^{(n)}, \hat{\mathbf{W}}_m^{(n)}$ are defined in section 3.1. And defining:*

$$(\hat{\Theta}_{i,m}^{(n)})^T = 2\pi j \xi_i \oplus \hat{\mathbf{W}}_m^{(n)}, \tag{29}$$

$$\hat{\Theta}_m = \big[ \hat{\Theta}_{0,m}^{(n)}, \hat{\Theta}_{1,m}^{(n)}, \dots, \hat{\Theta}_{N,m}^{(n)} \big], \tag{30}$$

$$\hat{\mathbf{X}}_m^{(n)} = \hat{\mathbf{A}}_m^{(n)} \hat{\mathbf{B}}_m^{(n)} \hat{\mathbf{C}}_m^{(n)}, \tag{31}$$

*with $\xi_i$ being the vector of frequencies for the $i$-dimension, $\oplus$ denoting element-wise product and $\hat{\mathbf{A}}_m^{(n)} \hat{\mathbf{B}}_m^{(n)} \hat{\mathbf{C}}_m^{(n)}$ being the SVD of $\hat{\mathbf{X}}_m^{(n)}$.*

*Proof.* See appendix C.1. □

# 4 EXPERIMENTS

## 4.1 SYNTHETIC DATA

We use the LRD model to generate matrices and tensors (eq. (3)). For two-dimensional matrices we use the filters learned from the city and fruit testing dataset from Zeiler et al. (2010). For three-dimensional tensors we use the filters learned on the basketball sequence used by Reixach (2023). In both cases we generate random activation-map matrices ($\{\mathbf{X}_m^{(n)}\}_{m=1,n=1}^{M,N}$) and learn the filters using the algorithm from Wohlberg (2017). For more details see appendix B.

For the matrix completion experiments, we evaluate the methods MF (Fan, 2022), FGSR (Fan et al., 2019), DMF (Fan & Cheng, 2018), M²DMTF (Fan, 2022) and LRD. And regarding the tensor completion experiments, we evaluate the methods FaLRTC (Liu et al., 2012), TenALS (Jain & Oh, 2014), TMac (Xu et al., 2015), KBR-TC (Xie et al., 2017), TRLRF (Yuan et al., 2019), OITNN-O (Wang et al., 2022), M²DMTF (Fan, 2022) and LRD (ours).

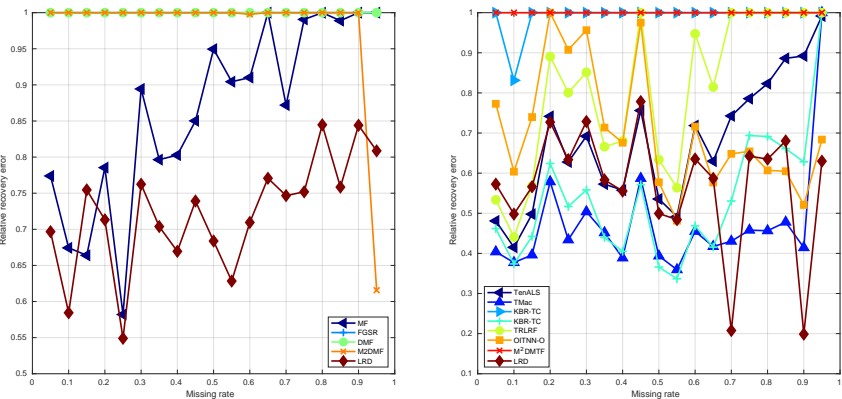

Figure 2: Performance evaluation of matrix and tensor completion on synthetic data ($\sigma = 1.0$).

Figure 2 shows the performance (average of 5 trials) with an Additive White Gaussian Noise (AWGN) with $\sigma = 1.0$ for both matrix and tensor completion. We use the relative recovery error (Fan et al., 2019) $\|\mathcal{P}_\Omega(\mathbf{Y}-\hat{\mathbf{Z}})\|/\|\mathcal{P}_\Omega(\mathbf{Y})\|$. We observe how our method systematically outperforms all the matrix completion methods considered. Regarding the tensor completion case our method is listed among the best achivers only surpassed by TMac and M²DMTF. For more results see appendix B.

## 4.2 REAL MATRICES

We consider a benchmark dataset: MovieLens-100k. It consists of a rating matrix (1 to 5) for movies given by users. The size (movies×users) of the rating matrix in the dataset is $1682 \times 943$ but due to computational limitations of our method we only consider a fourth part of the matrix with a size of $840 \times 460$. The matrix is highly incomplete, moreover we randomly mask out some of the known values with a certain missing ratio and add AWGN with a certain $\sigma$. The relative recovery errors

Table 1: Relative recovery error of matrix completion on MovieLens-100k.

| MR | $\sigma = 1.0$ | | | | | $\sigma = 2.0$ | | | | |
|---|---|---|---|---|---|---|---|---|---|---|
| | MF | FGSR | DMF | M²DMTF | LRD | MF | FGSR | DMF | M²DMTF | LRD |
| 10% | **0.2590** | 0.2957 | 0.3331 | 0.3248 | **0.2688** | **0.2664** | 0.4949 | 0.4586 | 0.4799 | **0.4463** |
| 30% | **0.2596** | 0.2904 | 0.3404 | 0.3343 | **0.2874** | **0.2769** | 0.4901 | 0.4803 | 0.5260 | **0.4500** |
| 50% | **0.2694** | **0.2835** | 0.3527 | 0.3521 | 0.2911 | **0.2917** | 0.4617 | 0.5129 | 0.6009 | **0.4184** |
| 70% | **0.2876** | **0.2881** | 0.3563 | 0.3688 | 0.2886 | **0.3198** | 0.4876 | 0.5610 | 0.6862 | **0.4404** |
| 90% | 0.3991 | 0.3520 | **0.3267** | **0.3385** | > 0.95 | **0.4512** | 0.5394 | **0.5218** | 0.5931 | > 0.95 |

Table 2: Relative recovery error of tensor completion on two real tensors ($\sigma = 1.0$ ).

| data | MR | FaLRTC | TenALS | TMac | KBR-TC | TRLRF | OITNN-O | M²DMTF | LRD |
|------|-----|--------|--------|------|--------|-------|---------|--------|-----|
| Amino | 10% | > 0.95 | > 0.95 | > 0.95 | > 0.95 | > 0.95 | > 0.95 | > 0.95 | **0.6459** |
| | 30% | > 0.95 | > 0.95 | > 0.95 | > 0.95 | > 0.95 | > 0.95 | > 0.95 | **0.6981** |
| | 50% | > 0.95 | > 0.95 | > 0.95 | > 0.95 | > 0.95 | > 0.95 | > 0.95 | **0.6851** |
| | 70% | > 0.95 | > 0.95 | > 0.95 | > 0.95 | > 0.95 | > 0.95 | > 0.95 | **0.6991** |
| | 90% | > 0.95 | > 0.95 | > 0.95 | > 0.95 | > 0.95 | > 0.95 | > 0.95 | **0.7356** |
| Flow | 10% | 0.8534 | **0.4583** | > 0.95 | > 0.95 | > 0.95 | > 0.95 | > 0.95 | **0.4825** |
| | 30% | 0.8923 | **0.4807** | > 0.95 | 0.9496 | > 0.95 | > 0.95 | > 0.95 | **0.4919** |
| | 50% | 0.9411 | **0.6049** | > 0.95 | > 0.95 | > 0.95 | > 0.95 | > 0.95 | **0.5064** |
| | 70% | > 0.95 | **0.6964** | > 0.95 | > 0.95 | > 0.95 | > 0.95 | > 0.95 | **0.5356** |
| | 90% | > 0.95 | > 0.95 | > 0.95 | > 0.95 | > 0.95 | > 0.95 | > 0.95 | **0.6094** |

for each case are reported in table 1 (average of 5 trials), where we have marked in blue and red color the best and second-best achievers. We can observe how MF is the best achiever and LRD (ours) is the second-best achiever in general. These results, together with the results reported in the appendix B where it can be seen how FGSR, DMF and M²DMTF perform well when less or no noise is considered makes us state that these methods do not perform well when noise is involved.

## 4.3 REAL TENSORS

We compare our method with the baselines of tensor completion on the following datasets: Amino acid fluorescence (Bro, 1997) ($5 \times 201 \times 61$) and Flow injection (Nørgaard & Ridder, 1994) ($12 \times 100 \times 89$) that we cut to the nearest even size in order to being able to compute the derivative component: ($4 \times 200 \times 60$) and ($12 \times 100 \times 88$) respectively. The relative recovery errors are reported in Tables 2 and 3 for an AWGN of $\sigma = 1.0$ and $\sigma = 2.0$ (average of 5 trials). In the appendix B, more results are reported. We observe how on the Amino dataset our method is the unique capable of reporting an acceptable error below $0.95$ while in the Flow dataset FaLRTC, TenALS and our method are the unique ones that are capable of reporting acceptable errors. When considering the results on the appendix we observe how our method is the unique that presents robustness against noise and a similar behaviour to the matrix completion case can be observed.

Table 3: Relative recovery error of tensor completion on two real tensors ($\sigma = 2.0$ ).

| data | MR | FalRTC | TenALS | TMac | KBR-TC | TRLRF | OITNN-O | M²DMTF | LRD |
|------|-----|--------|--------|------|--------|-------|---------|--------|-----|
| Amino | 10% | > 0.95 | > 0.95 | > 0.95 | > 0.95 | > 0.95 | > 0.95 | > 0.95 | **0.8450** |
| | 30% | > 0.95 | > 0.95 | > 0.95 | > 0.95 | > 0.95 | > 0.95 | > 0.95 | **0.8332** |
| | 50% | > 0.95 | > 0.95 | > 0.95 | > 0.95 | > 0.95 | > 0.95 | > 0.95 | **0.8572** |
| | 70% | > 0.95 | > 0.95 | > 0.95 | > 0.95 | > 0.95 | > 0.95 | > 0.95 | **0.8256** |
| | 90% | > 0.95 | > 0.95 | > 0.95 | > 0.95 | > 0.95 | > 0.95 | > 0.95 | **0.8993** |
| Flow | 10% | > 0.95 | **0.6460** | > 0.95 | > 0.95 | > 0.95 | > 0.95 | > 0.95 | **0.6554** |
| | 30% | > 0.95 | > 0.95 | > 0.95 | > 0.95 | > 0.95 | > 0.95 | > 0.95 | **0.6716** |
| | 50% | > 0.95 | > 0.95 | > 0.95 | > 0.95 | > 0.95 | > 0.95 | > 0.95 | **0.7021** |
| | 70% | > 0.95 | > 0.95 | > 0.95 | > 0.95 | > 0.95 | > 0.95 | > 0.95 | **0.7154** |
| | 90% | > 0.95 | > 0.95 | > 0.95 | > 0.95 | > 0.95 | > 0.95 | > 0.95 | **0.8241** |

## 5 CONCLUSION

This paper has proposed the use of LRD to the resolution of matrix and tensor completion problems. We have shown how LRD acts as a relaxation of the classical low-rank approach allowing for a greater number of solutions than the imposed by the low-rank constraint by performing a thorough theoretical analysis. We have also shown how the framework involved by LRD is flexible enough as it allows for easy algorithms to solve the learning process while at the same time facilitates the inclusion of priors such nuclear norm minimization or squared total variation regularization which allow the resolution of the problems considered as we have proved with the extensive empirical results presented.

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
