# A    OPTIMIZATION DETAILS

## A.1    CODE UPDATE STEP

Algorithm 2 and 3 give details about the resolution of equation (21) by using proposition 3.1.

---

**Algorithm 2 LRD-CODE.** Solves the LRD problem by means of an alternated approach for every $n$-mode. $\mathcal{L}(\mathbf{X}_m^{(n)})$ is defined in eq. (21). The optimization problem is solved by means of algorithm 3.

---
1: **Input:** $\mathcal{L}(\mathbf{X}_m^{(n)}), \{\mathbf{X}_{0,m}^{(n)}\}_{n=1,m=1}^{N,M}$
2: **Output:** $\{\mathbf{X}_m^{(n)}\}_{n=1,m=1}^{N,M}$
3: **while** not converged **do**
4:     **for** $n = 1$ **to** $N$ **do**
5:         **for** $m = 1$ **to** $M$ **do**
6:             $\mathbf{X}_m^{(n)} = \arg\min_{\mathbf{X}_m^{(n)}} \mathcal{L}(\mathbf{X}_m^{(n)})$
7:         **end for**
8:     **end for**
9: **end while**

---

**Algorithm 3 LRD-GD.** Solves the optimization problem in algorithm (2) by means of a gradient descent approach in the DFT domain, gradient given by eq. (28).

---
1: **Input:** $\nabla\mathcal{L}(\hat{\mathbf{x}}_m^{(n)}), \{\hat{\mathbf{x}}_{0,m}^{(n)}\}_{m=1}^{M}, T$
2: **Output:** $\{\hat{\mathbf{x}}_{T,m}^{(n)}\}_{m=1}^{M}$
3: **for** $t = 0$ **to** $T - 1$ **do**
4:     $\hat{\mathbf{x}}_{t+1,m}^{(n)} = \hat{\mathbf{x}}_{t,m}^{(n)} - \eta\nabla\mathcal{L}(\hat{\mathbf{x}}_{t,m}^{(n)})$
5: **end for**

---

## A.2    COMPUTATIONAL COMPLEXITY

The costly operations are the iterations of algorithm 2 which require to update $\hat{\mathbf{T}}_m^{(n)}$ that is an operation dominated by the computation of $\hat{\mathbf{F}}^{(n)}$ and $\hat{\mathbf{Q}}_m^{(n)}$. For a matrix completion case (two-dimensional) is

$$\mathcal{O}(2n^3 + Rn^2), \tag{32}$$

and for a three-dimensional tensor completion is

$$\mathcal{O}(n^5 + n^4 + (2+R)n^3). \tag{33}$$

## A.3    EXISTENCE OF SOLUTION

The existence of solution regarding the LRD decomposition is tied to the existence of the Kruskal tensor. It's existence is limited to $R < I_n$ for $n = 1, \cdots, N$.

# B    MORE ABOUT THE EXPERIMENTS

## B.1    SYNTHETIC DATA GENERATION

As explained in section 4 synthetic data is generated using the LRD model, that is

$$\mathcal{Z} = \sum_m \mathcal{D}_m * \mathcal{K}_m = \sum_m \mathcal{D}_m * [\![\mathbf{X}_m^{(1)}, \ldots, \mathbf{X}_m^{(N)}]\!], \tag{34}$$

where the dictionary $\{\mathcal{D}\}_{m=1}^M$ is the collection of filters learned and $\{\mathbf{X}_m^{(n)}\}_{m=1,n=1}^{M,N}$ the matrices that conform the kruskal activation map. For the two-dimensional case the dictionary is learned from the city and fruit testing dataset from Zeiler et al. (2010). For three-dimensional tensors we use the filters learned on the basketball sequence used by Reixach (2023). In both cases we generate random activation-map matrices drawn from the uniform distribution on the open interval $(0,1)$ and learn the filters using the algorithm from Wohlberg (2017). Figures 3 and 4 show the used filters.

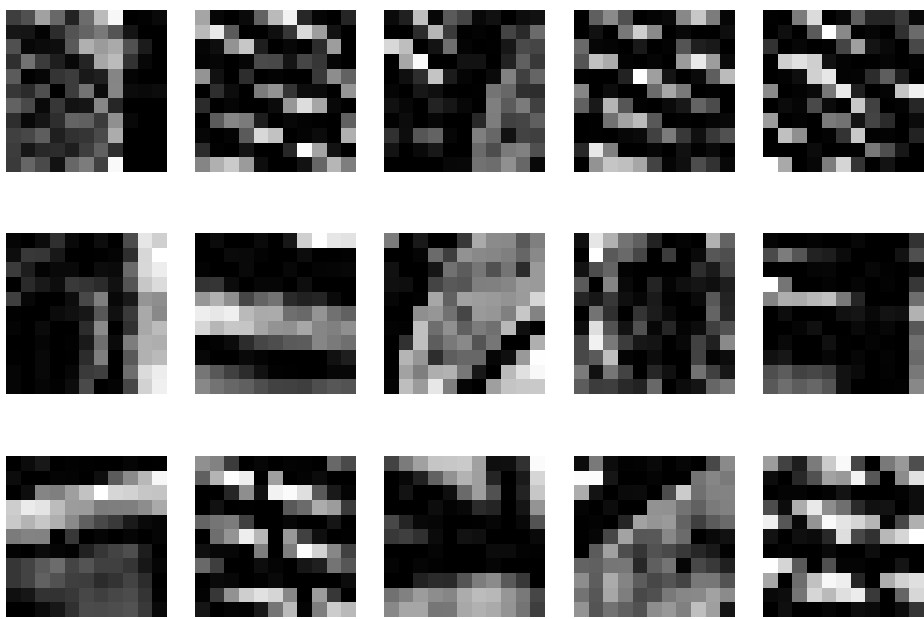

Figure 3: **Dictionary for the two-dimensional case.** From left to right and top to bottom the fifteen filters that conform the dictionary.

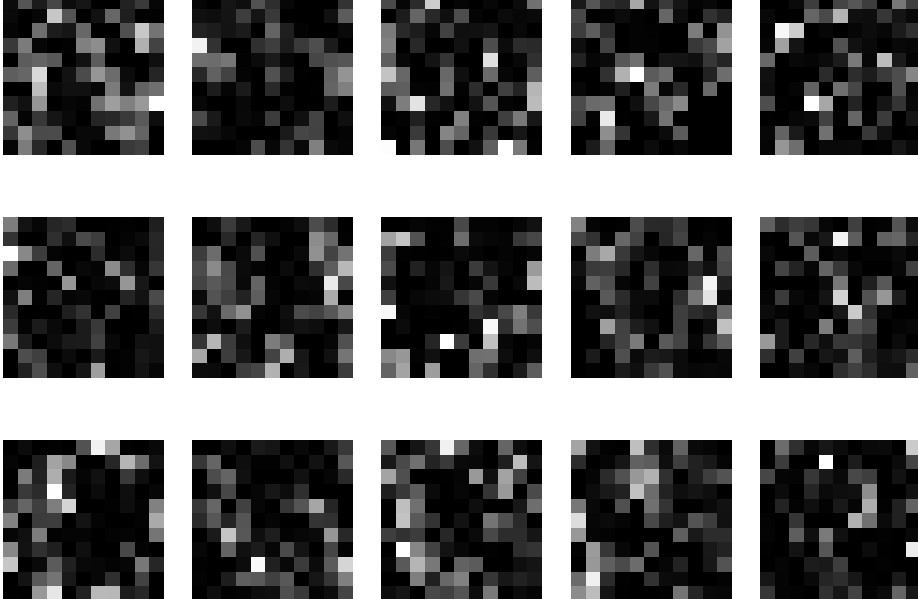

Figure 4: **Dictionary for the three-dimensional case.** From left to right and top to bottom the fifteen slides from each filter that conform the dictionary. As the filters are three-dimensional tensors here we picture the fifth slide in the third dimension of the filter.

## B.2 MORE RESULTS

Figures 5, 6 and 7 report synthetic results for AWGN with $\sigma = 0$, $\sigma = 0.1$ and $\sigma = 0.5$ respectively. We observe the behaviour described in section 4. Tables 4, 5, 6, 7, 8 and 9 report results on real

data for matrix and tensor completion. Regarding tensor completion, to the datasets considered in section 4 we also include SW-NIR kinetic data (Bijlsma & Smilde, 2000)($301 \times 241 \times 8$) that we cut to ($100 \times 100 \times 8$) due to computational limitations of our method. We observe the behaviour described in section 4.

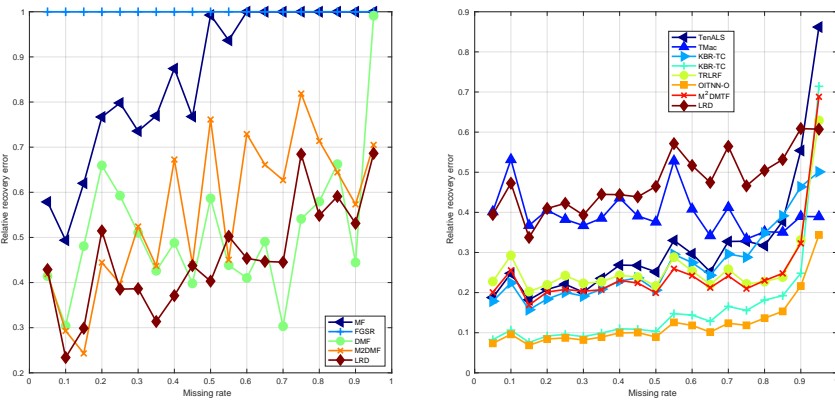

Figure 5: Performance evaluation of matrix and tensor completion on synthetic data ($\sigma = 0.0$).

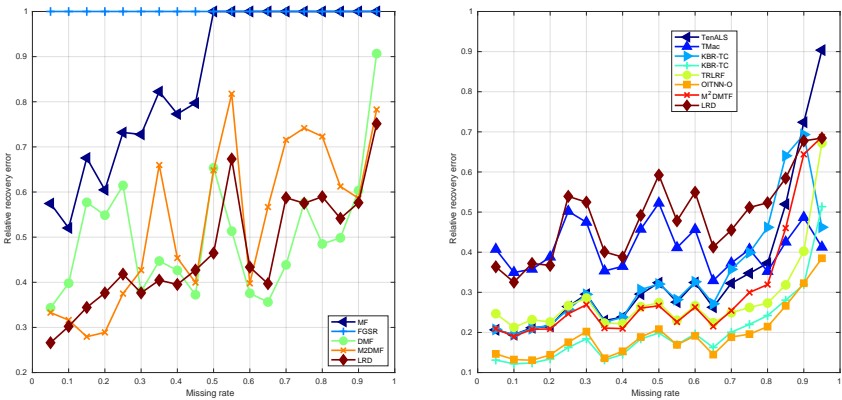

Figure 6: Performance evaluation of matrix and tensor completion on synthetic data ($\sigma = 0.1$).

Table 4: Relative recovery error of matrix completion on MovieLens-100k ($\sigma = 0.0$ ).

| Missing ratio | MF | FGSR | DMF | M$^2$DMTF | LRD |
|---|---|---|---|---|---|
| 10% | 0.2513 | **0.2413** | 0.2548 | **0.2454** | - |
| 30% | 0.2570 | **0.2480** | 0.2605 | **0.2529** | 0.2596 |
| 50% | 0.2615 | **0.2512** | 0.2621 | **0.2531** | 0.2608 |
| 70% | 0.2793 | **0.2648** | 0.2712 | **0.2546** | 0.2689 |
| 90% | 0.3913 | 0.3374 | **0.3157** | **0.2744** | 0.8056 |

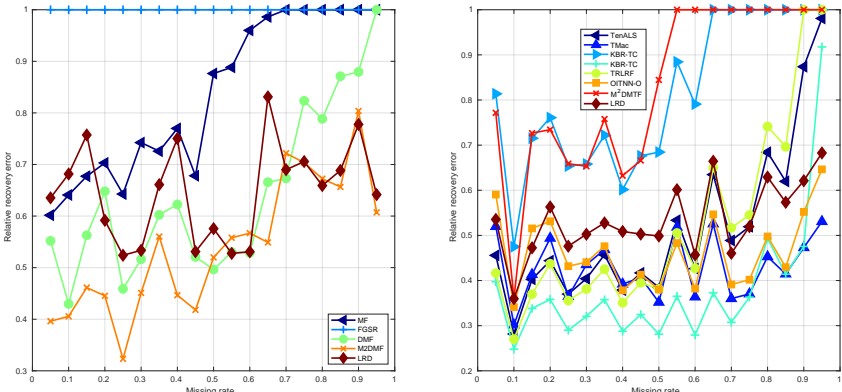

Figure 7: Performance evaluation of matrix and tensor completion on synthetic data ($\sigma = 0.5$).

Table 5: Relative recovery error of matrix completion on MovieLens-100k ($\sigma = 0.1$).

| Missing ratio | MF | FGSR | DMF | M$^2$DMTF | LRD |
|---|---|---|---|---|---|
| 10% | 0.2472 | **0.2376** | 0.2479 | **0.2450** | **0.2450** |
| 30% | 0.2531 | **0.2436** | 0.2567 | **0.2472** | 0.2547 |
| 50% | 0.2606 | **0.2495** | 0.2616 | **0.2543** | - |
| 70% | 0.2773 | 0.2621 | 0.2695 | **0.2586** | **0.2615** |
| 90% | 0.3892 | 0.3356 | **0.3170** | **0.2716** | 0.5842 |

Table 6: Relative recovery error of matrix completion on MovieLens-100k ($\sigma = 0.5$).

| Missing ratio | MF | FGSR | DMF | M$^2$DMTF | LRD |
|---|---|---|---|---|---|
| 10% | **0.2562** | **0.2484** | 0.2744 | 0.2600 | 0.2569 |
| 30% | **0.2585** | **0.2521** | 0.2804 | 0.2699 | - |
| 50% | **0.2605** | **0.2511** | 0.2781 | 0.2682 | 0.2770 |
| 70% | 0.2806 | **0.2654** | 0.2854 | 0.2786 | **0.2747** |
| 90% | 0.3999 | 0.3449 | **0.3221** | **0.2803** | 0.6660 |

### B.3 PARAMETER SETTING

#### B.3.1 PARAMETER SETTING ON SYNTHETIC DATA

- in MF, the factorization dimension $d$ is 5. The $\alpha$ is set to 1 and the maximum iteration is 2000.
- in FGSR, we use the variational form of Schatten-$1/2$ norm, $d = 30$, $\lambda = 0.015$. The maximum iteration is 2000.
- in DMF, the network structure is $[10, 50, 100, I_n]$. The weight decay parameters are set to 0.1 and the maximum iteration is 1000.
- in M$^2$DMTF (three-dimensional case), $L = 2$, $d_i = 10 \forall i$, $h_i = 50 \forall i$, $m_i = I_i \forall i$ and $\lambda_i = 1 \forall i$. The optimizer is iRprop+ and the maximum iteration is 2000.
- in LRD (two-dimensional case), $\alpha = 1 \cdot 10^{-16}$, $\gamma$ is choosen from $\{1 \cdot 10^{-4}, 1 \cdot 10^{-2}\}$, $\lambda = 0$, $r = 3$, $\eta = 1/50$ and $T = 500$. Maximum iteration is 100.
- In FaLRTC we set $\alpha_i = 1 \forall i$ and the maximum iteration is 200.

Table 7: Relative recovery error of tensor completion on three real tensors ($\sigma = 0.0$).

| data | MR | FalRTC | TenALS | TMac | KBR-TC | TRLRF | OITNN-O | M²DMTF | LRD |
|---|---|---|---|---|---|---|---|---|---|
| Amino | 10% | 0.0480 | 0.0197 | **0.0112** | 0.0186 | 0.0262 | **0.0115** | 0.0129 | 0.0441 |
| | 30% | 0.0472 | 0.0199 | **0.0116** | 0.0203 | 0.0454 | **0.0129** | 0.0137 | 0.0229 |
| | 50% | 0.0672 | 0.0203 | **0.0134** | 0.0197 | 0.0201 | 0.1360 | **0.0149** | 0.0380 |
| | 70% | 0.1124 | 0.0206 | **0.0169** | 0.0251 | 0.0216 | 0.0188 | **0.0157** | 0.0381 |
| | 90% | 0.01124 | 0.0221 | **0.0163** | 0.0251 | 0.0287 | 0.0188 | **0.0161** | 0.0504 |
| Flow | 10% | 0.0788 | 0.3939 | 0.0047 | **0.0014** | 0.0450 | **0.0021** | 0.0135 | 0.0127 |
| | 30% | 0.0998 | 0.3953 | 0.0057 | **0.0014** | 0.0460 | **0.0017** | 0.0150 | 0.0153 |
| | 50% | 0.1353 | 0.3958 | 0.0076 | **0.0016** | 0.0372 | **0.0021** | 0.0118 | 0.0136 |
| | 70% | 0.2121 | 0.3966 | 0.0103 | **0.0019** | 0.0234 | **0.0048** | 0.0142 | 0.0165 |
| | 90% | 0.5275 | 0.4005 | **0.0132** | **0.0039** | 0.0375 | 0.0480 | 0.0260 | 0.0721 |
| SW-NIR | 10% | 0.1285 | 0.1222 | **0.0000** | **0.0012** | 0.0055 | 0.0101 | 0.0057 | 0.0106 |
| | 30% | 0.1617 | 0.1131 | **0.0002** | > 0.95 | 0.0028 | **0.0010** | 0.0070 | 0.2030 |
| | 50% | 0.2913 | 0.1507 | 0.1404 | 0.1403 | 0.0436 | **0.0036** | **0.0059** | 0.5486 |
| | 70% | 0.6736 | > 0.95 | 0.2362 | 0.2363 | 0.0877 | **0.0062** | **0.0111** | > 0.95 |
| | 90% | 0.9413 | > 0.95 | 0.7050 | > 0.95 | 0.5046 | **0.4905** | **0.0459** | > 0.95 |

Table 8: Relative recovery error of tensor completion on three real tensors ($\sigma = 0.1$).

| data | MR | FalRTC | TenALS | TMac | KBR-TC | TRLRF | OITNN-O | M²DMTF | LRD |
|---|---|---|---|---|---|---|---|---|---|
| Amino | 10% | 0.1928 | **0.0854** | 0.2165 | 0.2634 | **0.1091** | 0.1554 | 0.2267 | 0.2782 |
| | 30% | 0.1898 | **0.0961** | 0.2513 | 0.2517 | **0.1555** | 0.1701 | 0.2574 | 0.2504 |
| | 50% | 0.1922 | **0.1121** | 0.3231 | 0.3080 | 0.1942 | 0.1858 | 0.3890 | **0.1816** |
| | 70% | 0.2285 | **0.1699** | 0.4667 | 0.2268 | **0.2040** | 0.2284 | 0.5782 | 0.3354 |
| | 90% | 0.2285 | **0.1587** | 0.4589 | **0.2268** | 0.2853 | 0.2284 | 0.5906 | 0.4117 |
| Flow | 10% | 0.1329 | 0.3977 | 0.2143 | 0.1115 | **0.1014** | 0.1928 | **0.0904** | 0.1455 |
| | 30% | 0.1488 | 0.3992 | 0.2290 | **0.1124** | **0.1128** | 0.1951 | 0.1116 | 0.1575 |
| | 50% | 0.1697 | 0.3966 | 0.2571 | 0.1523 | **0.1288** | 0.1903 | **0.1319** | 0.1761 |
| | 70% | 0.2237 | 0.3982 | 0.3367 | 0.2445 | 0.2099 | **0.2001** | **0.1804** | 0.2184 |
| | 90% | 0.5243 | 0.4068 | 0.6801 | **0.2524** | 0.3942 | **0.2876** | 0.4299 | 0.3288 |
| SW-NIR | 10% | 0.3469 | **0.1266** | 0.5085 | **0.1688** | 0.4394 | 0.4923 | 0.5717 | 0.4710 |
| | 30% | 0.4176 | **0.1380** | 0.5612 | **0.3251** | 0.6218 | 0.4850 | 0.4725 | 0.5481 |
| | 50% | 0.5336 | **0.1682** | 0.6547 | **0.2140** | 0.7099 | 0.4844 | 0.8324 | 0.4139 |
| | 70% | 0.7896 | > 0.95 | 0.9342 | **0.2903** | > 0.95 | **0.5755** | > 0.95 | 0.7776 |
| | 90% | > 0.95 | > 0.95 | > 0.95 | > 0.95 | > 0.95 | **0.8742** | > 0.95 | > 0.95 |

Table 9: Relative recovery error of tensor completion on three real tensors ($\sigma = 0.5$).

| data | MR | FalRTC | TenALS | TMac | KBR-TC | TRLRF | OITNN-O | M²DMTF | LRD |
|---|---|---|---|---|---|---|---|---|---|
| Amino | 10% | > 0.95 | **0.4625** | > 0.95 | > 0.95 | > 0.95 | **0.5992** | > 0.95 | 0.6037 |
| | 30% | > 0.95 | **0.5579** | > 0.95 | > 0.95 | > 0.95 | 0.6472 | > 0.95 | **0.4788** |
| | 50% | > 0.95 | **0.6784** | > 0.95 | > 0.95 | > 0.95 | 0.6982 | > 0.95 | **0.5671** |
| | 70% | > 0.95 | 0.9343 | > 0.95 | > 0.95 | > 0.95 | **0.7254** | > 0.95 | **0.6357** |
| | 90% | > 0.95 | > 0.95 | > 0.95 | > 0.95 | > 0.95 | **0.7254** | > 0.95 | **0.7310** |
| Flow | 10% | 0.5109 | **0.4069** | > 0.95 | 0.5551 | 0.7446 | 0.8524 | 0.7679 | **0.3731** |
| | 30% | 0.5365 | **0.4162** | > 0.95 | 0.5350 | 0.8835 | 0.8364 | > 0.95 | **0.3930** |
| | 50% | 0.5766 | **0.4232** | > 0.95 | 0.5712 | > 0.95 | 0.7671 | > 0.95 | **0.4138** |
| | 70% | 0.6765 | **0.4456** | > 0.95 | > 0.95 | > 0.95 | 0.7086 | > 0.95 | **0.4535** |
| | 90% | 0.9100 | **0.5281** | > 0.95 | > 0.95 | > 0.95 | 0.7164 | > 0.95 | **0.5263** |
| SW-NIR | 10% | > 0.95 | **0.6854** | > 0.95 | > 0.95 | > 0.95 | > 0.95 | > 0.95 | > 0.95 |
| | 30% | > 0.95 | **0.6938** | > 0.95 | > 0.95 | > 0.95 | > 0.95 | > 0.95 | > 0.95 |
| | 50% | > 0.95 | **0.8747** | > 0.95 | > 0.95 | > 0.95 | > 0.95 | > 0.95 | > 0.95 |
| | 70% | > 0.95 | > 0.95 | > 0.95 | > 0.95 | > 0.95 | > 0.95 | > 0.95 | > 0.95 |
| | 90% | > 0.95 | > 0.95 | > 0.95 | > 0.95 | > 0.95 | > 0.95 | > 0.95 | > 0.95 |

- in TenALS the initial rank is set to 1 and the maximum iteration is 20.

- in TMac, the rank is initialized to $[10, 10, 10]$ and adjusted adaptively. The maximum iteration is 1000.

- in KBR-TC, $\rho = 1.05$ and $\lambda = 0.01$. The maximum iteration is 300.

- in TRLRF the rank is set to $[3, 3, 3]$ and $\lambda = 10$. The maximum iteration is 500.
- in OITNN-O $\alpha = 1 \cdot 10^{-4}$. The maximum iteration is 300.
- in M$^2$DMTF (three-dimensional case), $L = 2$, $d_i = 10 \forall i$, $h_i = 20 \forall i$, $m_i = I_i \forall i$ and $\lambda_i = 1 \forall i$. The optimizer is iRprop+ and the maximum iteration is 3000.
- in LRD (three-dimensional case), $\alpha = 1 \cdot 10^{-16}$, $\gamma = 2 \cdot 10^{-5}$, $\lambda = 0$, $r = 3$, $\eta = 1/50$ and $T = 500$. Maximum iteration is 100.

### B.3.2 PARAMETER SETTING ON REAL DATA

- in MF, the factorization dimension $d$ is 5. The $\alpha$ is set to 1 and the maximum iteration is 2000.
- in FGSR, we use the variational form of Schatten-$1/2$ norm, $d = 50$, $\lambda = 0.015$. The maximum iteration is 2000.
- in DMF, the network structure is $[10, 50, 100, I_n]$. The weight decay parameters are set to 0.1 and the maximum iteration is 1000.
- in M$^2$DMTF (three-dimensional case), $L = 2$, $d_i = 10 \forall i$, $h_i = 50 \forall i$, $m_i = I_i \forall i$ and $\lambda_i = 1 \forall i$. The optimizer is iRprop+ and the maximum iteration is 2000.
- in LRD (two-dimensional case), $\alpha = 1 \cdot 10^{-10}$, $\gamma = 1 \cdot 10^{-8}$, $\lambda = 2$, $r = 5$, $\eta = 1/50$ and $T = 500$. Maximum iteration is 100.
- In FaLRTC we set $\alpha_i = 1 \forall i$ and the maximum iteration is 200.
- in TenALS the initial rank is set to 1 and the maximum iteration is 20.
- in TMac, the rank is initialized to $[10, 10, 10]$ and adjusted adaptively. The maximum iteration is 1000.
- in KBR-TC, $\rho = 1.05$ and $\lambda = 0.01$. The maximum iteration is 300.
- in TRLRF the rank is set to $[5, 5, 5]$ and $\lambda = 10$. The maximum iteration is 500.
- in OITNN-O $\alpha = 1 \cdot 10^{-4}$. The maximum iteration is 300.
- in M$^2$DMTF (three-dimensional case), $L = 2$, $d_i = 10 \forall i$, $h_i = 20 \forall i$, $m_i = I_i \forall i$ and $\lambda_i = 1 \forall i$. The optimizer is iRprop+ and the maximum iteration is 3000.
- in LRD (three-dimensional case), $\alpha = 1 \cdot 10^{-10}$, $\gamma$ is chosen from $\{1 \cdot 10^{-8}, 1 \cdot 10^{-6}, 1 \cdot 10^{-5}, 5 \cdot 10^{-5}\}$, $\lambda = 0$, $r = 4$, $\eta = 1/50$ and $T = 500$. Maximum iteration is 100.

## C PROOFS

### C.1 PROPOSITION 3.1

The squared total variation regularization is given by

$$\frac{\gamma}{2} \|\mathcal{U}\|_{TV}^2 = \frac{\gamma}{2} \|\nabla \mathcal{U}\|_2^2, \tag{35}$$

where,

$$\frac{\gamma}{2} \|\nabla \mathcal{U}\|_2^2 = \tag{36}$$

$$\frac{\gamma}{2} \left\| \left[ \left( \frac{\partial \mathbf{u}^{(n)}}{\partial \mathbf{t}_0} \right)^T, \left( \frac{\partial \mathbf{u}^{(n)}}{\partial \mathbf{t}_1} \right)^T, \dots, \left( \frac{\partial \mathbf{u}^{(n)}}{\partial \mathbf{t}_N} \right)^T \right]^T \right\|_2^2.$$

By using the derivative property of the DFT transform,

$$\mathcal{F}\left\{ \frac{\partial \mathbf{u}^{(n)}}{\partial \mathbf{t}_i} \right\} = 2\pi j \xi_i \oplus \mathcal{F}\{\mathbf{u}^{(n)}\} =$$

$$2\pi j \xi_i \oplus \hat{\mathbf{W}}^{(n)} \hat{\mathbf{x}}^{(n)}, \tag{37}$$

with $\xi_i$ and $\oplus$ defined in section 3.2. Together with the definition of $\hat{\Theta}^{(n)}$ leads us to the following expression,

$$\mathcal{F}\left\{\frac{\gamma}{2}\left\|\sum_{m=1}^{M}\boldsymbol{\mathcal{D}}_m * [\![\mathbf{X}_m^{(1)},\ldots,\mathbf{X}_m^{(N)}]\!]\right\|_{TV}^2\right\} =$$
$$\frac{\gamma}{2}\left\|(\hat{\Theta}^{(n)})^T\hat{\mathbf{x}}^{(n)}\right\|_2^2, \tag{38}$$

which can be derived in the following manner (complex derivative),

$$\frac{\partial \frac{\gamma}{2}\left\|(\hat{\Theta}^{(n)})^T\hat{\mathbf{x}}^{(n)}\right\|_2^2}{\partial(\hat{\mathbf{x}}^{(n)})^H} = \frac{\gamma}{2}(\hat{\Theta}^{(n)})^H\hat{\Theta}^{(n)}. \tag{39}$$

The result above combined with the solution of eq. (20) and the derivative of the nuclear norm operator brings us to eq. (28).

## C.2 THEOREM 2.2

This proof and the ones that follow are obtained following the method from Fan (2022), we also introduce the following lemma from the same work (Lemma 1):

**Lemma C.1.** *Let $S$ be a set defined over tensors of size $I_1 \times I_2 \times \cdots \times I_n$. Let $|S|$ be the $e$-covering number of $S$ w.r.t. the Frobenius norm. Let $I_\pi = \prod_{i=1}^{n} I_i$. Suppose $\boldsymbol{\mathcal{Z}} \in S$ and $\max(\|\boldsymbol{\mathcal{Y}}\|_\infty, \|\boldsymbol{\mathcal{Z}}\|_\infty) \le \xi$. Then with probability at least $1 - 2I_\pi^{-1}$:*

$$\sup_{\boldsymbol{\mathcal{Z}}\in S}\left|\frac{1}{\sqrt{I_\pi}}\|\boldsymbol{\mathcal{Y}} - \boldsymbol{\mathcal{Z}}\|_F - \frac{1}{\sqrt{|\Omega|}}\|P_\Omega(\boldsymbol{\mathcal{Y}} - \boldsymbol{\mathcal{Z}})\|_F\right| \le \frac{2\epsilon}{\sqrt{|\Omega|}} + \left(\frac{8\xi^4\log(|S|I_\pi)}{|\Omega|}\right)^{1/4} \tag{40}$$

*Proof.* See Fan (2022) appendix D.1. $\qquad\qquad\square$

After, we introduce a new lemma that shows an upper bound for the covering number of the Low-rank Deconvolution matrix set:

**Lemma C.2.** *Let $S = \{\mathbf{Z} \in \mathbb{R}^{I_2 \times I_1}: \mathbf{Z} = \sum_{m=1}^{M}\mathbf{F}_2 {}^{(2)}\hat{\mathbf{D}}_m \oplus \hat{\mathbf{X}}_m^{(2)}(\hat{\mathbf{X}}_m^{(1)})^T\mathbf{F}_1, \|\hat{\mathbf{X}}_m^{(n)}\|_F \le \beta_m^n, \|{}^{(2)}\hat{\mathbf{D}}_m\|_F \le \beta_m^0, m = 1,\ldots,M, n = 1,2\}$ where $\hat{\mathbf{X}}_m^{(n)} \in \mathbb{C}^{I_n \times r}$, $\mathbf{F}_1 \in \mathbb{C}^{I_1 \times I_1}$ and $\mathbf{F}_2 \in \mathbb{C}^{I_2 \times I_2}$ are the inverse DFT matrices and ${}^{(2)}\hat{\mathbf{D}}_m \in \mathbb{C}^{I_2 \times I_1}$. Then the covering number of $S$ w.r.t. the Frobenius norm satisfy:*

$$\mathcal{N}(S, \|\cdot\|_F, \epsilon) \le \left(\frac{3^4 M^3 \sum_{m=1}^{M}(\beta_m^0\beta_m^1\beta_m^2)^3}{\epsilon}\right)^{I_1 r + 2I_\pi} \tag{41}$$

*Proof.* See appendix C.6 $\qquad\qquad\square$

With Lemma C.1 and Lemma C.2 we obtain

$$\frac{1}{\sqrt{I_\pi}}\|\mathbf{Z} - \hat{\mathbf{Z}}\|_F = \frac{1}{\sqrt{I_\pi}}\|\mathbf{Y} - \mathbf{E} - \hat{\mathbf{Z}}\|_F$$

$$\le \frac{1}{\sqrt{I_\pi}}\|\mathbf{Y} - \hat{\mathbf{Z}}\|_F + \frac{1}{\sqrt{I_\pi}}\|\mathbf{E}\|_F$$

$$\le \frac{1}{\sqrt{|\Omega|}}\|P_\Omega(\mathbf{Y} - \hat{\mathbf{Z}})\|_F + \frac{1}{\sqrt{I_\pi}}\|\mathbf{E}\|_F + \frac{2\epsilon}{\sqrt{|\Omega|}} + \left(\frac{8\xi^4\log(|S|I_\pi)}{|\Omega|}\right)^{1/4}$$

$$\le \frac{1}{\sqrt{|\Omega|}}\|P_\Omega(\mathbf{Y} - \hat{\mathbf{Z}})\|_F + \frac{1}{\sqrt{I_\pi}}\|\mathbf{E}\|_F + \frac{2\epsilon}{\sqrt{|\Omega|}} +$$

$$\left(\frac{8\xi^4(\log(I_\pi) + (I_1 r + 2I_\pi)\log(\frac{3^4 M^3 \sum_{m=1}^{M}(\beta_m^0\beta_m^1\beta_m^2)^3}{\epsilon}))}{|\Omega|}\right)^{1/4} \tag{42}$$

## C.3 THEOREM 2.4

This is a special case of proof C.2 where $^{(2)}\hat{\mathbf{D}} = \mathbf{1}$ and DFT matrices are not longer needed as we consider spatial versions $\mathbf{X}^{(n)}$. With that we introduce a new lemma that shows an upper bound for the covering number of the classical Low-rank matrix set:

**Lemma C.3.** *Let* $S = \{\mathbf{Z} \in \mathbb{R}^{I_2 \times I_1} \colon \mathbf{Z} = \mathbf{X}^{(2)}(\mathbf{X}^{(1)})^T, \|\mathbf{X}^{(n)}\|_F \leq \beta^n, n = 1, 2\}$ *where* $\mathbf{X}^{(n)} \in \mathbb{R}^{I_n \times r}$. *Then the covering number of* $S$ *w.r.t. the Frobenius norm satisfy:*

$$\mathcal{N}(S, \|\cdot\|_F, \epsilon') \leq \left( \frac{12(\beta^1 \beta^2)^2}{\epsilon'} \right)^{I_1 r + I_\pi} \tag{43}$$

*Proof.* See appendix C.7 □

With Lemma C.1 and Lemma C.3 we obtain

$$\begin{aligned}
\frac{1}{\sqrt{I_\pi}} \|\mathbf{Z} - \hat{\mathbf{Z}}\|_F &= \frac{1}{\sqrt{I_\pi}} \|\mathbf{Y} - \mathbf{E} - \hat{\mathbf{Z}}\|_F \\
&\leq \frac{1}{\sqrt{I_\pi}} \|\mathbf{Y} - \hat{\mathbf{Z}}\|_F + \frac{1}{\sqrt{I_\pi}} \|\mathbf{E}\|_F \\
&\leq \frac{1}{\sqrt{|\Omega|}} \|P_\Omega(\mathbf{Y} - \hat{\mathbf{Z}})\|_F + \frac{1}{\sqrt{I_\pi}} \|\mathbf{E}\|_F + \frac{2\epsilon'}{\sqrt{|\Omega|}} + \left( \frac{8\xi^4 \log(|S|I_\pi)}{|\Omega|} \right)^{1/4} \\
&\leq \frac{1}{\sqrt{|\Omega|}} \|P_\Omega(\mathbf{Y} - \hat{\mathbf{Z}})\|_F + \frac{1}{\sqrt{I_\pi}} \|\mathbf{E}\|_F + \frac{2\epsilon'}{\sqrt{|\Omega|}} + \\
&\quad \left( \frac{8\xi^4 (\log(I_\pi) + (I_1 r + I_\pi) \log(\frac{12(\beta^1 \beta^2)^2}{\epsilon'}))}{|\Omega|} \right)^{1/4}
\end{aligned} \tag{44}$$

## C.4 THEOREM 2.6

This is the three-dimensional variation of proof C.2. To that end we introduce a new lemma that shows an upper bound for the covering number of the three-dimensional Low-rank Deconvolution matrix set:

**Lemma C.4.** *Let* $S = \{\mathbf{Z} \in \mathbb{R}^{I_3 \times I_1 I_2} \colon \mathbf{Z} = \sum_{m=1}^{M} \mathbf{F}_3 \,^{(3)}\hat{\mathbf{D}}_m \oplus \hat{\mathbf{X}}_m^{(3)} (\hat{\mathbf{X}}_m^{(1)} \odot \hat{\mathbf{X}}_m^{(2)})^T \mathbf{F}_{12}, \|\hat{\mathbf{X}}_m^{(n)}\|_F \leq \beta_m^n, \|^{(3)}\hat{\mathbf{D}}_m\|_F \leq \beta_m^0, m = 1, \dots, M, n = 1, 2, 3\}$ *where* $\hat{\mathbf{X}}_m^{(n)} \in \mathbb{C}^{I_n \times r}$, $\mathbf{F}_{12} \in \mathbb{C}^{I_1 I_2 \times I_1 I_2}$ *and* $\mathbf{F}_3 \in \mathbb{C}^{I_3 \times I_3}$ *are the inverse DFT matrices and* $^{(3)}\hat{\mathbf{D}}_m \in \mathbb{C}^{I_3 \times I_1 I_2}$. *Then the covering number of* $S$ *w.r.t. the Frobenius norm satisfy:*

$$\mathcal{N}(S, \|\cdot\|_F, \epsilon) \leq \left( \frac{3 \cdot 4^4 M^4 \sum_{m=1}^{M} (\beta_m^0 \beta_m^1 \beta_m^2 \beta_m^3 k_r)^4}{\epsilon} \right)^{(1+I_2) I_1 r + 2 I_\pi} \tag{45}$$

*Proof.* See appendix C.8 □

With Lemma C.1 and Lemma C.4 we obtain

$$
\begin{aligned}
\frac{1}{\sqrt{I_\pi}}\|\mathbf{Z} - \hat{\mathbf{Z}}\|_F &= \frac{1}{\sqrt{I_\pi}}\|\mathbf{Y} - \mathbf{E} - \hat{\mathbf{Z}}\|_F \\
&\leq \frac{1}{\sqrt{I_\pi}}\|\mathbf{Y} - \hat{\mathbf{Z}}\|_F + \frac{1}{\sqrt{I_\pi}}\|\mathbf{E}\|_F \\
&\leq \frac{1}{\sqrt{|\Omega|}}\|P_\Omega(\mathbf{Y} - \hat{\mathbf{Z}})\|_F + \frac{1}{\sqrt{I_\pi}}\|\mathbf{E}\|_F + \frac{2\epsilon}{\sqrt{|\Omega|}} + \left(\frac{8\xi^4 \log(|S|I_\pi)}{|\Omega|}\right)^{1/4} \\
&\leq \frac{1}{\sqrt{|\Omega|}}\|P_\Omega(\mathbf{Y} - \hat{\mathbf{Z}})\|_F + \frac{1}{\sqrt{I_\pi}}\|\mathbf{E}\|_F + \frac{2\epsilon}{\sqrt{|\Omega|}} + \\
&\quad \left(\frac{8\xi^4(\log(I_\pi) + ((1 + I_2)I_1 r + 2I_\pi)\log(\frac{3\cdot 4^4 M^4 \sum_{m=1}^M (\beta_m^0 \beta_m^1 \beta_m^2 \beta_m^3 k_r)^4}{\epsilon}))}{|\Omega|}\right)^{1/4}
\end{aligned}
\tag{46}
$$

## C.5 THEOREM 2.8

This is a special case of proof C.4 where ${}^{(2)}\hat{\mathbf{D}} = \mathbf{1}$ and DFT matrices are not longer needed as we consider spatial versions $\mathbf{X}^{(n)}$. With that we introduce a new lemma that shows an upper bound for the covering number of the classical three-dimensional Low-rank matrix set:

**Lemma C.5.** *Let* $S = \{\mathbf{Z} \in \mathbb{R}^{I_3 \times I_1 I_2}: \mathbf{Z} = \mathbf{X}^{(3)}(\mathbf{X}^{(1)} \odot \mathbf{X}^{(2)})^T, \|\mathbf{X}^{(n)}\|_F \leq \beta^n, n = 1, 2, 3\}$ *where* $\mathbf{X}^{(n)} \in \mathbb{R}^{I_n \times r}$. *Then the covering number of* $S$ *w.r.t. the Frobenius norm satisfy:*

$$
\mathcal{N}(S, \|\cdot\|_F, \epsilon') \leq \left(\frac{3^4(\beta^1 \beta^2 \beta^3 k_r)^3}{\epsilon'}\right)^{(1+I_2)I_1 r + I_\pi}
\tag{47}
$$

*Proof.* See appendix C.9 □

With Lemma C.1 and Lemma C.5 we obtain

$$
\begin{aligned}
\frac{1}{\sqrt{I_\pi}}\|\mathbf{Z} - \hat{\mathbf{Z}}\|_F &= \frac{1}{\sqrt{I_\pi}}\|\mathbf{Y} - \mathbf{E} - \hat{\mathbf{Z}}\|_F \\
&\leq \frac{1}{\sqrt{I_\pi}}\|\mathbf{Y} - \hat{\mathbf{Z}}\|_F + \frac{1}{\sqrt{I_\pi}}\|\mathbf{E}\|_F \\
&\leq \frac{1}{\sqrt{|\Omega|}}\|P_\Omega(\mathbf{Y} - \hat{\mathbf{Z}})\|_F + \frac{1}{\sqrt{I_\pi}}\|\mathbf{E}\|_F + \frac{2\epsilon'}{\sqrt{|\Omega|}} + \left(\frac{8\xi^4 \log(|S|I_\pi)}{|\Omega|}\right)^{1/4} \\
&\leq \frac{1}{\sqrt{|\Omega|}}\|P_\Omega(\mathbf{Y} - \hat{\mathbf{Z}})\|_F + \frac{1}{\sqrt{I_\pi}}\|\mathbf{E}\|_F + \frac{2\epsilon'}{\sqrt{|\Omega|}} + \\
&\quad \left(\frac{8\xi^4(\log(I_\pi) + ((1 + I_2)I_1 r + I_\pi)\log(\frac{3^4(\beta^1 \beta^2 \beta^3 k_r)^3}{\epsilon'}))}{|\Omega|}\right)^{1/4}
\end{aligned}
\tag{48}
$$

## C.6 LEMMA C.2

Let $\mathbf{Z} = \sum_{m=1}^M \mathbf{F}_2 {}^{(2)}\hat{\mathbf{D}}_m \oplus \hat{\mathbf{X}}_m^{(2)}(\hat{\mathbf{X}}_m^{(1)})^T \mathbf{F}_1$ where $\hat{\mathbf{X}}_m^{(n)} \in \mathbb{C}^{I_n \times r}$, $\mathbf{F}_1 \in \mathbb{C}^{I_1 \times I_1}$ and $\mathbf{F}_2 \in \mathbb{C}^{I_2 \times I_2}$ are the inverse DFT matrices and $\hat{\mathbf{D}}_m^{(2)} \in \mathbb{C}^{I_2 \times I_1}$. We give the following two lemmas:

**Lemma C.6.** *Let* $S_{ab} := \{\hat{\mathbf{A}} \in \mathbb{C}^{a \times b}: \hat{\mathbf{A}} = \mathbf{F}_n \mathbf{A}, \mathbf{A} \in \mathbb{R}^{a \times b}, \|\mathbf{A}\|_F \leq \beta, \mathbf{F}_2 \in \mathbb{C}^{a \times a}$ *the inverse DFT matrix,* $\|\mathbf{F}_n\|_F = 1\}$. *Then there exist an* $\epsilon$-*net* $\tilde{S}_{ab}$ *obeying*

$$
\mathcal{N}(S_{ab}, \|\cdot\|_F, \epsilon) \leq \left(\frac{3\beta}{\epsilon}\right)^{ab}
\tag{49}
$$

*such that* $\left\|\hat{\mathbf{A}} - \tilde{\mathbf{A}}\right\|_F \leq \epsilon$.

*Proof.* See appendix C.10 □

**Lemma C.7.** *Let* $\hat{\mathbf{A}} \in \mathbb{C}^{a \times b}$ *and* $\hat{\mathbf{B}} \in \mathbb{C}^{a \times b}$ *then*

$$\left\| \hat{\mathbf{A}} \oplus \hat{\mathbf{B}} \right\|_F \leq tr(\hat{\mathbf{A}}\hat{\mathbf{B}}^T) \leq \left\| \hat{\mathbf{A}} \right\|_F \left\| \hat{\mathbf{B}} \right\|_F \tag{50}$$

*Where* $\oplus$ *is defined above and depicts hadamard product.*

Now replace $\epsilon$ with $\epsilon/\zeta_m^n$ and let $\left\| \hat{\mathbf{X}}_m^{(n)} - \tilde{\mathbf{X}}_m^{(n)} \right\|_F \leq \frac{\epsilon}{\zeta_m^n}, m = 1, \ldots, M, \quad n = 1, 2, \quad \|^{(2)}\hat{\mathbf{D}}_m - {}^{(2)}\tilde{\mathbf{D}}_m\|_F \leq \frac{\epsilon}{\zeta_m^0}$. Let $\zeta_m^0 = 3M\beta_m^1\beta_m^2, \zeta_m^1 = 3M\beta_m^0\beta_m^2$ and $\zeta_m^2 = 3M\beta_m^0\beta_m^1$.

$$
\begin{aligned}
\left\| \mathbf{Z} - \bar{\mathbf{Z}} \right\|_F &= \left\| \sum_{m=1}^M \mathbf{F}_2 {}^{(2)}\hat{\mathbf{D}}_m \oplus \hat{\mathbf{X}}_m^{(2)}(\hat{\mathbf{X}}_m^{(1)})^T \mathbf{F}_1 - \sum_{m=1}^M \mathbf{F}_2 {}^{(2)}\tilde{\mathbf{D}}_m \oplus \tilde{\mathbf{X}}_m^{(2)}(\tilde{\mathbf{X}}_m^{(1)})^T \mathbf{F}_1 \right\|_F \\
&= \left\| \sum_{m=1}^M \mathbf{F}_2 {}^{(2)}\hat{\mathbf{D}}_m \oplus \hat{\mathbf{X}}_m^{(2)}(\hat{\mathbf{X}}_m^{(1)})^T \mathbf{F}_1 \pm \sum_{m=1}^M \mathbf{F}_2 {}^{(2)}\tilde{\mathbf{D}}_m \oplus \hat{\mathbf{X}}_m^{(2)}(\hat{\mathbf{X}}_m^{(1)})^T \mathbf{F}_1 \right. \\
&\quad \left. \pm \sum_{m=1}^M \mathbf{F}_2 {}^{(2)}\tilde{\mathbf{D}}_m \oplus \tilde{\mathbf{X}}_m^{(2)}(\hat{\mathbf{X}}_m^{(1)})^T \mathbf{F}_1 - \sum_{m=1}^M \mathbf{F}_2 {}^{(2)}\tilde{\mathbf{D}}_m \oplus \tilde{\mathbf{X}}_m^{(2)}(\tilde{\mathbf{X}}_m^{(1)})^T \mathbf{F}_1 \right\|_F \\
&\leq \left\| \sum_{m=1}^M \mathbf{F}_2 {}^{(2)}\hat{\mathbf{D}}_m \oplus \hat{\mathbf{X}}_m^{(2)}(\hat{\mathbf{X}}_m^{(1)})^T \mathbf{F}_1 - \sum_{m=1}^M \mathbf{F}_2 {}^{(2)}\tilde{\mathbf{D}}_m \oplus \hat{\mathbf{X}}_m^{(2)}(\hat{\mathbf{X}}_m^{(1)})^T \mathbf{F}_1 \right\|_F \\
&\quad + \left\| \sum_{m=1}^M \mathbf{F}_2 {}^{(2)}\tilde{\mathbf{D}}_m \oplus \hat{\mathbf{X}}_m^{(2)}(\hat{\mathbf{X}}_m^{(1)})^T \mathbf{F}_1 - \sum_{m=1}^M \mathbf{F}_2 {}^{(2)}\tilde{\mathbf{D}}_m \oplus \tilde{\mathbf{X}}_m^{(2)}(\hat{\mathbf{X}}_m^{(1)})^T \mathbf{F}_1 \right\|_F \\
&\quad + \left\| \sum_{m=1}^M \mathbf{F}_2 {}^{(2)}\tilde{\mathbf{D}}_m \oplus \tilde{\mathbf{X}}_m^{(2)}(\hat{\mathbf{X}}_m^{(1)})^T \mathbf{F}_1 - \sum_{m=1}^M \mathbf{F}_2 {}^{(2)}\tilde{\mathbf{D}}_m \oplus \tilde{\mathbf{X}}_m^{(2)}(\tilde{\mathbf{X}}_m^{(1)})^T \mathbf{F}_1 \right\|_F \\
&\leq \sum_{m=1}^M \left\| \mathbf{F}_2 \right\|_F \left\| {}^{(2)}\hat{\mathbf{D}}_m - {}^{(2)}\tilde{\mathbf{D}}_m \right\|_F \left\| \hat{\mathbf{X}}_m^{(2)} \right\|_F \left\| \hat{\mathbf{X}}_m^{(1)} \right\|_F \left\| \mathbf{F}_1 \right\|_F \\
&\quad + \sum_{m=1}^M \left\| \mathbf{F}_2 \right\|_F \left\| {}^{(2)}\tilde{\mathbf{D}}_m \right\|_F \left\| \hat{\mathbf{X}}_m^{(2)} - \tilde{\mathbf{X}}_m^{(2)} \right\|_F \left\| \hat{\mathbf{X}}_m^{(1)} \right\|_F \left\| \mathbf{F}_1 \right\|_F \\
&\quad + \sum_{m=1}^M \left\| \mathbf{F}_2 \right\|_F \left\| {}^{(2)}\tilde{\mathbf{D}}_m \right\|_F \left\| \tilde{\mathbf{X}}_m^{(2)} \right\|_F \left\| \hat{\mathbf{X}}_m^{(1)} - \tilde{\mathbf{X}}_m^{(1)} \right\|_F \left\| \mathbf{F}_1 \right\|_F \\
&\leq \sum_{m=1}^M \frac{\epsilon}{\zeta_m^0} \beta_m^1 \beta_m^2 + \sum_{m=1}^M \frac{\epsilon}{\zeta_m^2} \beta_m^0 \beta_m^1 + \sum_{m=1}^M \frac{\epsilon}{\zeta_m^1} \beta_m^0 \beta_m^2 \\
&= \epsilon.
\end{aligned} \tag{51}
$$

The second inequality utilized the submultiplicativity of the Frobenius norm and Lemma C.7. Therefore, $\bar{S}$ is an $\epsilon$-cover of $S$. Then, we have

$$
\begin{aligned}
\mathcal{N}(S, \|\cdot\|_F, \epsilon) &\leq \sum_{m=1}^M \prod_{n=0}^2 \left( \frac{3\beta_m^n \zeta_m^n}{\epsilon} \right)^{I_1 r + 2I_\pi} \\
&= \sum_{m=1}^M \left( \frac{3^4 M^3 (\beta_m^0 \beta_m^1 \beta_m^2)^3}{\epsilon} \right)^{I_1 r + 2I_\pi} \\
&= \left( \frac{3^4 M^3 \sum_{m=1}^M (\beta_m^0 \beta_m^1 \beta_m^2)^3}{\epsilon} \right)^{I_1 r + 2I_\pi}
\end{aligned} \tag{52}
$$

## C.7   LEMMA C.3

Let $\mathbf{Z} = \mathbf{X}^{(2)}(\mathbf{X}^{(1)})^T$ where $\mathbf{X}^{(n)} \in \mathbb{C}^{I_n \times r}$. Using Lemma C.6, replace $\epsilon$ with $\epsilon'/\bar{\zeta}^n$ and let $\left\|\mathbf{X}^{(n)} - \tilde{\mathbf{X}}^{(n)}\right\|_F \le \frac{\epsilon}{\bar{\zeta}^n}, n = 1, 2$. Let $\bar{\zeta}^1 = 2\beta^2$ and $\bar{\zeta}^2 = 2\beta^1$.

$$
\begin{aligned}
\left\|\mathbf{Z} - \bar{\mathbf{Z}}\right\|_F &= \left\|\mathbf{X}^{(2)}(\mathbf{X}^{(1)})^T - \bar{\mathbf{X}}^{(2)}(\bar{\mathbf{X}}^{(1)})^T\right\|_F \\
&= \left\|\mathbf{X}^{(2)}(\mathbf{X}^{(1)})^T \pm \bar{\mathbf{X}}^{(2)}(\mathbf{X}^{(1)})^T - \bar{\mathbf{X}}^{(2)}(\bar{\mathbf{X}}^{(1)})^T\right\|_F \\
&\le \left\|\mathbf{X}^{(2)}(\mathbf{X}^{(1)})^T - \bar{\mathbf{X}}^{(2)}(\mathbf{X}^{(1)})^T\right\|_F + \left\|\bar{\mathbf{X}}^{(2)}(\mathbf{X}^{(1)})^T - \bar{\mathbf{X}}^{(2)}(\bar{\mathbf{X}}^{(1)})^T\right\|_F \\
&\le \left\|\mathbf{X}_m^{(2)} - \bar{\mathbf{X}}_m^{(2)}\right\|_F \left\|\mathbf{X}_m^{(1)}\right\|_F + \left\|\bar{\mathbf{X}}_m^{(2)}\right\|_F \left\|\mathbf{X}_m^{(1)} - \bar{\mathbf{X}}_m^{(1)}\right\|_F \\
&\le \frac{\epsilon'}{\bar{\zeta}^2}\beta^1 + \frac{\epsilon'}{\bar{\zeta}^1}\beta^2 = \epsilon'.
\end{aligned}
\tag{53}
$$

The second inequality utilized the submultiplicativity of the Frobenius norm. Therefore, $\bar{S}$ is an $\epsilon'$-cover of $S$. Then, we have

$$
\begin{aligned}
\mathcal{N}(S, \|\cdot\|_F, \epsilon') &\le \prod_{n=0}^{1}\left(\frac{3\beta^n \bar{\zeta}^n}{\epsilon'}\right)^{I_1 r + I_\pi} \\
&= \left(\frac{3 \cdot 2^2 (\beta^1 \beta^2)^2}{\epsilon'}\right)^{I_1 r + I_\pi} = \left(\frac{12(\beta^1 \beta^2)^2}{\epsilon'}\right)^{I_1 r + I_\pi}.
\end{aligned}
\tag{54}
$$

## C.8   LEMMA C.4

Let $\mathbf{Z} = \sum_{m=1}^{M} \mathbf{F}_3 {}^{(3)}\hat{\mathbf{D}}_m \oplus \hat{\mathbf{X}}_m^{(3)}(\hat{\mathbf{X}}_m^{(1)} \odot \hat{\mathbf{X}}_m^{(2)})^T \mathbf{F}_{12}$ where $\hat{\mathbf{X}}_m^{(n)} \in \mathbb{C}^{I_n \times r}$, $\mathbf{F}_{12} \in \mathbb{C}^{I_1 I_2 \times I_1 I_2}$ and $\mathbf{F}_3 \in \mathbb{C}^{I_3 \times I_3}$ are the inverse DFT matrices and $\hat{\mathbf{D}}_m^{(3)} \in \mathbb{C}^{I_3 \times I_1 I_2}$. We give the following lemma:

**Lemma C.8.** *Let $\hat{\mathbf{A}} \in \mathbb{C}^{a \times b}$ and $\hat{\mathbf{B}} \in \mathbb{C}^{a \times b}$ then*

$$
\left\|\hat{\mathbf{A}} \odot \hat{\mathbf{B}}\right\|_F = \left\|\hat{\mathbf{A}}\right\|_F \left\|\hat{\mathbf{B}}\right\|_F \left\|\hat{\mathbf{I}}_r\right\|_F
\tag{55}
$$

*Where $\odot$ is defined above and depicts khatri-rao product.*

Now replace $\epsilon$ with $\epsilon/\zeta_m^n$ and let $\left\|\hat{\mathbf{X}}_m^{(n)} - \tilde{\mathbf{X}}_m^{(n)}\right\|_F \le \frac{\epsilon}{\zeta_m^n}, m = 1, \ldots, M, \quad n = 1, 2, \quad \|{}^{(2)}\hat{\mathbf{D}}_m - {}^{(2)}\tilde{\mathbf{D}}_m\|_F \le \frac{\epsilon}{\zeta_m^0}$. Let $\zeta_m^0 = 4M\beta_m^1\beta_m^2\beta_m^3 k_r$, $\zeta_m^1 = 4M\beta_m^0\beta_m^2\beta_m^3 k_r$, $\zeta_m^2 = 4M\beta_m^0\beta_m^1\beta_m^3 k_r$ and

$$\zeta_m^3 = 4M\beta_m^0\beta_m^1\beta_m^2 k_r.$$

$$
\begin{aligned}
\left\|\mathbf{Z} - \bar{\mathbf{Z}}\right\|_F &= \left\|\sum_{m=1}^{M} \mathbf{F}_3{}^{(3)}\hat{\mathbf{D}}_m \oplus \hat{\mathbf{X}}_m^{(3)}(\hat{\mathbf{X}}_m^{(1)} \odot \hat{\mathbf{X}}_m^{(2)})^T\mathbf{F}_{12} - \sum_{m=1}^{M} \mathbf{F}_3{}^{(3)}\tilde{\mathbf{D}}_m \oplus \tilde{\mathbf{X}}_m^{(3)}(\tilde{\mathbf{X}}_m^{(1)} \odot \tilde{\mathbf{X}}_m^{(2)})^T\mathbf{F}_{12}\right\|_F \\
&= \left\|\sum_{m=1}^{M} \mathbf{F}_3{}^{(3)}\hat{\mathbf{D}}_m \oplus \hat{\mathbf{X}}_m^{(3)}(\hat{\mathbf{X}}_m^{(1)} \odot \hat{\mathbf{X}}_m^{(2)})^T\mathbf{F}_{12} \pm \sum_{m=1}^{M} \mathbf{F}_3{}^{(3)}\tilde{\mathbf{D}}_m \oplus \hat{\mathbf{X}}_m^{(3)}(\hat{\mathbf{X}}_m^{(1)} \odot \hat{\mathbf{X}}_m^{(2)})^T\mathbf{F}_{12}\right. \\
&\quad \pm \sum_{m=1}^{M} \mathbf{F}_3{}^{(3)}\tilde{\mathbf{D}}_m \oplus \tilde{\mathbf{X}}_m^{(3)}(\hat{\mathbf{X}}_m^{(1)} \odot \hat{\mathbf{X}}_m^{(2)})^T\mathbf{F}_{12} \pm \sum_{m=1}^{M} \mathbf{F}_3{}^{(3)}\tilde{\mathbf{D}}_m \oplus \tilde{\mathbf{X}}_m^{(3)}(\tilde{\mathbf{X}}_m^{(1)} \odot \hat{\mathbf{X}}_m^{(2)})^T\mathbf{F}_{12} \\
&\quad \left. - \sum_{m=1}^{M} \mathbf{F}_3{}^{(3)}\tilde{\mathbf{D}}_m \oplus \tilde{\mathbf{X}}_m^{(3)}(\tilde{\mathbf{X}}_m^{(1)} \odot \tilde{\mathbf{X}}_m^{(2)})^T\mathbf{F}_{12}\right\|_F \\
&\leq \left\|\sum_{m=1}^{M} \mathbf{F}_3{}^{(3)}\hat{\mathbf{D}}_m \oplus \hat{\mathbf{X}}_m^{(3)}(\hat{\mathbf{X}}_m^{(1)} \odot \hat{\mathbf{X}}_m^{(2)})^T\mathbf{F}_{12} - \sum_{m=1}^{M} \mathbf{F}_3{}^{(3)}\tilde{\mathbf{D}}_m \oplus \hat{\mathbf{X}}_m^{(3)}(\hat{\mathbf{X}}_m^{(1)} \odot \hat{\mathbf{X}}_m^{(2)})^T\mathbf{F}_{12}\right\|_F \\
&\quad + \left\|\sum_{m=1}^{M} \mathbf{F}_3{}^{(3)}\tilde{\mathbf{D}}_m \oplus \hat{\mathbf{X}}_m^{(3)}(\hat{\mathbf{X}}_m^{(1)} \odot \hat{\mathbf{X}}_m^{(2)})^T\mathbf{F}_{12} - \sum_{m=1}^{M} \mathbf{F}_3{}^{(3)}\tilde{\mathbf{D}}_m \oplus \tilde{\mathbf{X}}_m^{(3)}(\hat{\mathbf{X}}_m^{(1)} \odot \hat{\mathbf{X}}_m^{(2)})^T\mathbf{F}_{12}\right\|_F \\
&\quad + \left\|\sum_{m=1}^{M} \mathbf{F}_3{}^{(3)}\tilde{\mathbf{D}}_m \oplus \tilde{\mathbf{X}}_m^{(3)}(\hat{\mathbf{X}}_m^{(1)} \odot \hat{\mathbf{X}}_m^{(2)})^T\mathbf{F}_{12} - \sum_{m=1}^{M} \mathbf{F}_3{}^{(3)}\tilde{\mathbf{D}}_m \oplus \tilde{\mathbf{X}}_m^{(3)}(\tilde{\mathbf{X}}_m^{(1)} \odot \hat{\mathbf{X}}_m^{(2)})^T\mathbf{F}_{12}\right\|_F \\
&\quad + \left\|\sum_{m=1}^{M} \mathbf{F}_3{}^{(3)}\tilde{\mathbf{D}}_m \oplus \tilde{\mathbf{X}}_m^{(3)}(\tilde{\mathbf{X}}_m^{(1)} \odot \hat{\mathbf{X}}_m^{(2)})^T\mathbf{F}_{12} - \sum_{m=1}^{M} \mathbf{F}_3{}^{(3)}\tilde{\mathbf{D}}_m \oplus \tilde{\mathbf{X}}_m^{(3)}(\tilde{\mathbf{X}}_m^{(1)} \odot \tilde{\mathbf{X}}_m^{(2)})^T\mathbf{F}_{12}\right\|_F \\
&\leq \sum_{m=1}^{M} \left\|\mathbf{F}_3\right\|_F \left\|{}^{(3)}\hat{\mathbf{D}}_m - {}^{(3)}\tilde{\mathbf{D}}_m\right\|_F \left\|\hat{\mathbf{X}}_m^{(3)}\right\|_F \left\|\hat{\mathbf{X}}_m^{(1)}\right\|_F \left\|\hat{\mathbf{X}}_m^{(2)}\right\|_F \left\|\mathbf{I}_r\right\|_F \left\|\mathbf{F}_{12}\right\|_F \\
&\quad + \sum_{m=1}^{M} \left\|\mathbf{F}_3\right\|_F \left\|{}^{(3)}\tilde{\mathbf{D}}_m\right\|_F \left\|\hat{\mathbf{X}}_m^{(3)} - \tilde{\mathbf{X}}_m^{(3)}\right\|_F \left\|\hat{\mathbf{X}}_m^{(1)}\right\|_F \left\|\hat{\mathbf{X}}_m^{(2)}\right\|_F \left\|\mathbf{I}_r\right\|_F \left\|\mathbf{F}_{12}\right\|_F \\
&\quad + \sum_{m=1}^{M} \left\|\mathbf{F}_3\right\|_F \left\|{}^{(3)}\tilde{\mathbf{D}}_m\right\|_F \left\|\tilde{\mathbf{X}}_m^{(3)}\right\|_F \left\|\hat{\mathbf{X}}_m^{(1)} - \tilde{\mathbf{X}}_m^{(1)}\right\|_F \left\|\hat{\mathbf{X}}_m^{(2)}\right\|_F \left\|\mathbf{I}_r\right\|_F \left\|\mathbf{F}_{12}\right\|_F \\
&\quad + \sum_{m=1}^{M} \left\|\mathbf{F}_3\right\|_F \left\|{}^{(3)}\tilde{\mathbf{D}}_m\right\|_F \left\|\tilde{\mathbf{X}}_m^{(3)}\right\|_F \left\|\tilde{\mathbf{X}}_m^{(1)}\right\|_F \left\|\hat{\mathbf{X}}_m^{(2)} - \tilde{\mathbf{X}}_m^{(2)}\right\|_F \left\|\mathbf{I}_r\right\|_F \left\|\mathbf{F}_{12}\right\|_F \\
&\leq \sum_{m=1}^{M} \frac{\epsilon}{\zeta_m^0}\beta_m^1\beta_m^2\beta_m^3 k_r + \sum_{m=1}^{M} \frac{\epsilon}{\zeta_m^3}\beta_m^0\beta_m^1\beta_m^2 k_r + \sum_{m=1}^{M} \frac{\epsilon}{\zeta_m^1}\beta_m^0\beta_m^2\beta_m^3 k_r + \sum_{m=1}^{M} \frac{\epsilon}{\zeta_m^3}\beta_m^0\beta_m^1\beta_m^3 k_r \\
&= \epsilon.
\end{aligned}
\tag{56}
$$

The second inequality utilized the submultiplicativity of the Frobenius norm and Lemma C.8. Therefore, $\bar{S}$ is an $\epsilon$-cover of $S$. Then, we have

$$
\begin{aligned}
\mathcal{N}(S, \left\|\cdot\right\|_F, \epsilon) &\leq \sum_{m=1}^{M} \prod_{n=0}^{3} \left(\frac{3\beta_m^n\zeta_m^n}{\epsilon}\right)^{(1+I_2)I_1 r + 2I_\pi} \\
&= \sum_{m=1}^{M} \left(\frac{3 \cdot 4^4 M^4 (\beta_m^0\beta_m^1\beta_m^2\beta_m^3 k_r)^4}{\epsilon}\right)^{(1+I_2)I_1 r + 2I_\pi} \\
&= \left(\frac{3 \cdot 4^4 M^4 \sum_{m=1}^{M} (\beta_m^0\beta_m^1\beta_m^2\beta_m^3 k_r)^4}{\epsilon}\right)^{(1+I_2)I_1 r + 2I_\pi}
\end{aligned}
\tag{57}
$$

## C.9 LEMMA C.5

Let $\mathbf{Z} = \mathbf{X}^{(3)}(\mathbf{X}^{(1)} \odot \mathbf{X}^{(2)})^T$ where $\mathbf{X}^{(n)} \in \mathbb{C}^{I_n \times r}$. Using Lemma C.6, replace $\epsilon$ with $\epsilon'/\bar{\zeta}^n$ and let $\left\| \mathbf{X}^{(n)} - \tilde{\mathbf{X}}^{(n)} \right\|_F \leq \frac{\epsilon}{\bar{\zeta}^n}, n = 1, 2, 3$. Let $\bar{\zeta}^1 = 3\beta^2\beta^3$, $\bar{\zeta}^2 = 3\beta^1\beta^3$ and $\bar{\zeta}^3 = 3\beta^1\beta^2$.

$$
\begin{aligned}
\left\| \mathbf{Z} - \bar{\mathbf{Z}} \right\|_F &= \left\| \mathbf{X}^{(3)}(\mathbf{X}^{(1)} \odot \mathbf{X}^{(2)})^T - \bar{\mathbf{X}}^{(3)}(\bar{\mathbf{X}}^{(1)} \odot \bar{\mathbf{X}}^{(2)})^T \right\|_F \\
&= \left\| \mathbf{X}^{(3)}(\mathbf{X}^{(1)} \odot \mathbf{X}^{(2)})^T \pm \bar{\mathbf{X}}^{(3)}(\mathbf{X}^{(1)} \odot \mathbf{X}^{(2)})^T \pm \bar{\mathbf{X}}^{(3)}(\bar{\mathbf{X}}^{(1)} \odot \mathbf{X}^{(2)})^T - \bar{\mathbf{X}}^{(3)}(\bar{\mathbf{X}}^{(1)} \odot \bar{\mathbf{X}}^{(2)})^T \right\|_F \\
&\leq \left\| \mathbf{X}^{(3)}(\mathbf{X}^{(1)} \odot \mathbf{X}^{(2)})^T - \bar{\mathbf{X}}^{(3)}(\mathbf{X}^{(1)} \odot \mathbf{X}^{(2)})^T \right\|_F + \left\| \bar{\mathbf{X}}^{(3)}(\mathbf{X}^{(1)} \odot \mathbf{X}^{(2)})^T - \bar{\mathbf{X}}^{(3)}(\bar{\mathbf{X}}^{(1)} \odot \mathbf{X}^{(2)})^T \right\|_F \\
&\quad + \left\| \bar{\mathbf{X}}^{(3)}(\bar{\mathbf{X}}^{(1)} \odot \mathbf{X}^{(2)})^T - \bar{\mathbf{X}}^{(3)}(\bar{\mathbf{X}}^{(1)} \odot \bar{\mathbf{X}}^{(2)})^T \right\|_F \\
&\leq \left\| \mathbf{X}_m^{(3)} - \bar{\mathbf{X}}_m^{(3)} \right\|_F \left\| \mathbf{X}_m^{(1)} \right\|_F \left\| \mathbf{X}_m^{(2)} \right\|_F + \left\| \bar{\mathbf{X}}_m^{(3)} \right\|_F \left\| \mathbf{X}_m^{(1)} - \bar{\mathbf{X}}_m^{(1)} \right\|_F \left\| \mathbf{X}_m^{(2)} \right\|_F \\
&\quad + \left\| \bar{\mathbf{X}}_m^{(3)} \right\|_F \left\| \bar{\mathbf{X}}_m^{(1)} \right\|_F \left\| \mathbf{X}_m^{(2)} - \bar{\mathbf{X}}_m^{(2)} \right\|_F \\
&\leq \frac{\epsilon'}{\bar{\zeta}^3}\beta^1\beta^2 + \frac{\epsilon'}{\bar{\zeta}^1}\beta^3\beta^2 + \frac{\epsilon'}{\bar{\zeta}^2}\beta^3\beta^1 = \epsilon'. \quad (58)
\end{aligned}
$$

The second inequality utilized the submultiplicativity of the Frobenius norm and Lemma C.8. Therefore, $\bar{S}$ is an $\epsilon'$-cover of $S$. Then, we have

$$
\begin{aligned}
\mathcal{N}(S, \|\cdot\|_F, \epsilon') &\leq \prod_{n=0}^{2} \left( \frac{3\beta^n \bar{\zeta}^n}{\epsilon'} \right)^{(1+I_2)I_1 r + I_\pi} \\
&= \left( \frac{3 \cdot 3^3 (\beta^1\beta^2\beta^3)^3}{\epsilon'} \right)^{(1+I_2)I_1 r + I_\pi} = \left( \frac{3^4 (\beta^1\beta^2\beta^3)^3}{\epsilon'} \right)^{(1+I_2)I_1 r + I_\pi}. \quad (59)
\end{aligned}
$$

## C.10 LEMMA C.6

$$
\left\| \hat{\mathbf{A}} - \tilde{\mathbf{A}} \right\|_F = \left\| \mathbf{F}_n \mathbf{A} - \mathbf{F}_n \bar{\mathbf{A}} \right\|_F \leq \left\| \mathbf{A} - \bar{\mathbf{A}} \right\|_F. \quad (60)
$$

The inequality utilized the submultiplicativity of the Frobenius norm.