# OpenReview forum: "Matrix and Tensor Completion with Noise via Low-rank Deconvolution"
_ICLR.cc/2024/Conference — ICLR 2024 Conference Withdrawn Submission_

### Official Review · Reviewer_1xgP · 2023-10-29

**Soundness:** 1 poor
**Presentation:** 1 poor
**Contribution:** 1 poor
**Rating:** 1
**Confidence:** 4

**Summary:**

This paper proposes utilizing Low-Rank Deconvolution (LRD) to solve the matrix and tensor completion problems, and contains extensive mathematical parts, aiming to prove that LRD can act as a relaxation of the classical low-rank approach. Experiments are conducted on synthetic data, MovieLens-100k, Amino acid fluorescence, and Flow injection datasets, benchmarking with tensor completion algorithms such as FalRTC, TenALS, TMac, etc.

**Strengths:**

Analytic forms in Section 2 of LRD vs LR.

**Weaknesses:**

1. The tensor completion algorithms used in this work (FaLRTC, TMac, TenALS, etc.) are quite old, newer TC algorithms abound in past few years and should be included. Perhaps more importantly, as stated in 4.2, "due to computational limitations .... we only consider ... 840 x 460". First, you are missing a complexity analysis of your algorithm, then also such a small size is far from being practical in any real-world scenarios!

2. In 4.1, I was expecting image & video completion examples measured by PSNR and runtimes. These are de facto experiments for mainstream tensor completion papers. You should add these examples (e.g. the popular Kodak24) for the paper to meet the bar for evaluation.

3. Recent years have shown coordinate networks (in the context of implicit neural representation or INR) to be superb for image/video completion, e.g., the famous SIREN. Your work should be compared with INR also.

4. Deconvolution or convolution? Also, your LRD completion algorithm is based on existing algorithms, mainly following Reixach (2023), what exactly are the innovations here?

5. After reading the paper, I am curious about what are the research gaps in the matrix and tensor completion problems using the existing algorithms? What are the necessity and motivations for using LRD in this research area? In fact, Table 1 demonstrates your LRD isn't as good as simple MF.

Editorial:
Abstract and conclusion, "than the imposed" should be "than those imposed".

Lots of "Eq. equation #", please fix.

The J in (9) has a different font from the J at 2 lines above.

**Questions:**

See weaknesses for questions.

**Details Of Ethics Concerns:**

Nil

---

### Official Review · Reviewer_PNwi · 2023-10-29

**Soundness:** 3 good
**Presentation:** 2 fair
**Contribution:** 4 excellent
**Rating:** 6
**Confidence:** 3

**Summary:**

In this paper the authors consider a matrix and tensor completion method based on the ansatz as a convolution of a dictionary and an activation map. In the presence of random noise with known variance, several theorems are proved about the recovery accuracy which is achieved with certain probability. Thus, the new method is compared with classical low-rank completion. In the experimental section several experiments are given, more of them in the appendix, many of this experiments performed on synthetic data or on artificially noisy data rather than on real-world problems.

**Strengths:**

* theoretical evaluation for the proposed method has been carried out

* the problem of matrix and tensor complеtion is actual, so the works with new methods and their theoretical justification are very valuable

**Weaknesses:**

* no code provided;
* experiments are rather few, and mostly conducted on synthetic data, or noise is artificially added to them;
* in experiments LPD shows quite contradictory results, there is no strict outperformance of competitors;
* the narrative is non-linear, there are too frequent references to sections that go further in the text, and without them some places are difficult to understand in detail;
 * The formulations of the theorems are not neat, for example, the parameters $\beta_i$, which are present in the formulations of theorems 2.2, 2.4, 2.6, 2.8, are not defined in the main text.

**Questions:**

- How parameter $\sigma$ affect the estimates of Theorems 2.2, 2.4, 2.6, 2.8?
- What does "complexity" on figure 1 means?
- Why recover error of the right-hand side of fig. 2 does not tend to upper values as missing rate grows?
- The authors say that" Regarding the tensor completion case our method is listed among the best achivers only surpassed by TMac and M2DMTF" but according to the right-hand side of fig. 2 the least error is shown by TMac, KBR-TC and new method.

Minor:

- it is not clear what  symbol star "$\star$" in (3)--(4) means (it is defined only on p.6).
-  figure 1, the right side of the $x$ axis, "100" is cut to "10"
- typo: p3 2nd paragraph  "use **an** special"; further in this paragraph suggestion: put brackets $(\prod_tI_t)/I_n$
- typo: p6 ("Eq." and "equation") Algorithm 1 LRD. Solves Eq. equation 20

---

### Official Review · Reviewer_BDxW · 2023-11-06

**Soundness:** 3 good
**Presentation:** 2 fair
**Contribution:** 2 fair
**Rating:** 3
**Confidence:** 3

**Summary:**

The paper presents the application of the low-rank-deconvolution model on the matrix and tensor completion problem. The authors build the connection between the low-rank-deconvolution model and the classic low-rank model. They also provide several experiments showcasing the efficacy of their methods.

**Strengths:**

1. The experiment result seems promising. The model is tested on both synthetic and real datasets.

**Weaknesses:**

1. The paper is not well-presented. There are several typos and missing notations. For example, the equal sign in Theorem 2.2, 2.4, 2.6, 2.8 should all be "$\leq$". In theorem 2.2 the authors start with Suppose ... are given by (12), what does this mean? Are those variables the minimizer of (12)? Also, important definitions for notations like "$|| \cdot||_\infty$" are missing. The authors spent the whole paper comparing with the LR method but didn't provide a rigorous formulation for LR. Instead, they just present its matrix version as formulation (1). In section 2, the authors keep referring the readers to section 3 for the definition of DFT domains, but fail to explain it with enough clarity in section 3. Similar problems like the above make it difficult for readers to follow.

2. There are very few discussions about the connection between this paper and prior arts. Considering how popular matrix completion is, at least a section of related works should be present.

3. The theoretical result is not very strong and seems repetitive. It seems to me that section 2.1 and 2.2 are presenting almost the same result under dimension 2 and 3. The authors are welcome to provide more details to help readers contextualize their theoretical contribution. The current discussion seems insufficient.

**Questions:**

1. In (4), the regularizer for X is of Frobenius norm instead of nuclear norm. Since we are solving a low-rank matrix completion problem, can you explain why?